# FastGRNN: A Fast, Accurate, Stable and Tiny Kilobyte Sized Gated Recurrent Neural Network

**Aditya Kusupati**[†]**, Manish Singh**[§]**, Kush Bhatia**[‡]**,**
**Ashish Kumar**[‡]**, Prateek Jain**[†] **and Manik Varma**[†]
[†]Microsoft Research India
[§]Indian Institute of Technology Delhi
[‡]University of California Berkeley
{t-vekusu,prajain,manik}@microsoft.com, singhmanishiitd@gmail.com
kush@cs.berkeley.edu, ashish_kumar@berkeley.edu

## Abstract

This paper develops the FastRNN and FastGRNN algorithms to address the twin RNN limitations of inaccurate training and inefficient prediction. Previous approaches have improved accuracy at the expense of prediction costs making them infeasible for resource-constrained and real-time applications. Unitary RNNs have increased accuracy somewhat by restricting the range of the state transition matrix's singular values but have also increased the model size as they require a larger number of hidden units to make up for the loss in expressive power. Gated RNNs have obtained state-of-the-art accuracies by adding extra parameters thereby resulting in even larger models. FastRNN addresses these limitations by adding a residual connection that does not constrain the range of the singular values explicitly and has only two extra scalar parameters. FastGRNN then extends the residual connection to a gate by reusing the RNN matrices to match state-of-the-art gated RNN accuracies but with a 2-4x smaller model. Enforcing FastGRNN's matrices to be low-rank, sparse and quantized resulted in accurate models that could be up to 35x smaller than leading gated and unitary RNNs. This allowed FastGRNN to accurately recognize the "Hey Cortana" wakeword with a 1 KB model and to be deployed on severely resource-constrained IoT microcontrollers too tiny to store other RNN models. FastGRNN's code is available at [30].

## 1 Introduction

**Objective**: This paper develops the FastGRNN (an acronym for a Fast, Accurate, Stable and Tiny Gated Recurrent Neural Network) algorithm to address the twin RNN limitations of inaccurate training and inefficient prediction. FastGRNN almost matches the accuracies and training times of state-of-the-art unitary and gated RNNs but has significantly lower prediction costs with models ranging from 1 to 6 Kilobytes for real-world applications.

**RNN training and prediction**: It is well recognized that RNN training is inaccurate and unstable as non-unitary hidden state transition matrices could lead to exploding and vanishing gradients for long input sequences and time series. An equally important concern for resource-constrained and real-time applications is the RNN's model size and prediction time. Squeezing the RNN model and code into a few Kilobytes could allow RNNs to be deployed on billions of Internet of Things (IoT) endpoints having just 2 KB RAM and 32 KB flash memory [17, 29]. Similarly, squeezing the RNN model and code into a few Kilobytes of the 32 KB L1 cache of a Raspberry Pi or smartphone, could significantly reduce the prediction time and energy consumption and make RNNs feasible for real-

time applications such as wake word detection [27, 11, 12, 42, 43], predictive maintenance [46, 1], human activity recognition [3, 2], *etc*.

**Unitary and gated RNNs**: A number of techniques have been proposed to stabilize RNN training based on improved optimization algorithms [40, 26], unitary RNNs [5, 24, 37, 47, 50, 54, 25] and gated RNNs [20, 13, 14]. While such approaches have increased the RNN prediction accuracy they have also significantly increased the model size. Unitary RNNs have avoided gradients exploding and vanishing by limiting the range of the singular values of the hidden state transition matrix. This has led to only limited gains in prediction accuracy as the optimal transition matrix might often not be close to unitary. Unitary RNNs have compensated by learning higher dimensional representations but, unfortunately, this has led to larger model sizes. Gated RNNs [20, 13, 14] have stabilized training by adding extra parameters leading to state-of-the-art prediction accuracies but with models that might sometimes be even larger than unitary RNNs.

**FastRNN**: This paper demonstrates that standard RNN training could be stabilized with the addition of a residual connection [19, 44, 22, 7] having just 2 additional scalar parameters. Residual connections for RNNs have been proposed in [22] and further studied in [7]. This paper proposes the FastRNN architecture and establishes that a simple variant of [22, 7] with learnt weighted residual connections (2) can lead to provably stable training and near state-of-the-art prediction accuracies with lower prediction costs than all unitary and gated RNNs. In particular, FastRNN's prediction accuracies could be: (a) up to 19% higher than a standard RNN; (b) could often surpass the accuracies of all unitary RNNs and (c) could be just shy of the accuracies of leading gated RNNs. FastRNN's empirical performance could be understood on the basis of theorems proving that for an input sequence with $T$ steps and appropriate setting of residual connection weights: (a) FastRNN converges to a stationary point within $O(1/\epsilon^2)$ SGD iterations (see Theorem 3.1), independent of $T$, while the *same analysis* for a standard RNN reveals an upper bound of $O(2^T)$ iterations and (b) FastRNN's generalization error bound is independent of $T$ whereas the *same proof technique* reveals an exponential bound for standard RNNs.

**FastGRNN**: Inspired by this analysis, this paper develops the novel FastGRNN architecture by converting the residual connection to a gate while reusing the RNN matrices. This allowed FastGRNN to match, and sometimes exceed, state-of-the-art prediction accuracies of LSTM, GRU, UGRNN and other leading gated RNN techniques while having 2-4x fewer parameters. Enforcing FastGRNN's matrices to be low-rank, sparse and quantized led to a minor decrease in the prediction accuracy but resulted in models that could be up to 35x smaller and fit in 1-6 Kilobytes for many applications. For instance, using a 1 KB model, FastGRNN could match the prediction accuracies of all other RNNs at the task of recognizing the "Hey Cortana" wakeword. This allowed FastGRNN to be deployed on IoT endpoints, such as the Arduino Uno, which were too small to hold other RNN models. On slightly larger endpoints, such as the Arduino MKR1000 or Due, FastGRNN was found to be 18-42x faster at making predictions than other leading RNN methods.

**Contributions**: This paper makes two contributions. First, it rigorously studies the residual connection based FastRNN architecture which could often outperform unitary RNNs in terms of training time, prediction accuracy and prediction cost. Second, inspired by FastRNN, it develops the Fast-GRNN architecture which could almost match state-of-the-art accuracies and training times but with prediction costs that could be lower by an order of magnitude. FastRNN and FastGRNN's code can be downloaded from [30].

## 2   Related Work

**Residual connections**: Residual connections have been studied extensively in CNNs [19, 44] as well as RNNs [22, 7]. The Leaky Integration Unit architecture [22] proposed residual connections for RNNs but were unable to learn the state transition matrix due to the problem of exploding and vanishing gradients. They therefore sampled the state transition matrix from a hand-crafted distribution with spectral radius less than one. This limitation was addressed in [7] where the state transition matrix was learnt but the residual connections were applied to only a few hidden units and with randomly sampled weights. Unfortunately, the distribution from which the weights were sampled could lead to an ill-conditioned optimization problem. In contrast, the FastRNN architecture leads to provably stable training with just two learnt weights connected to all the hidden units.

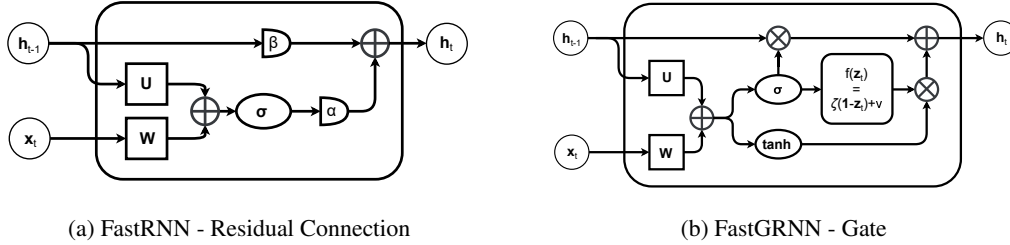

(a) FastRNN - Residual Connection          (b) FastGRNN - Gate

Figure 1: Block diagrams for FastRNN (a) and FastGRNN (b). FastGRNN uses shared matrices $\mathbf{W}$, $\mathbf{U}$ to compute both the hidden state $\mathbf{h}_t$ as well as the gate $\mathbf{z}_t$.

**Unitary RNNs**: Unitary RNNs [5, 50, 37, 24, 47, 25] stabilize RNN training by learning only well-conditioned state transition matrices. This limits their expressive power and prediction accuracy while increasing training time. For instance, SpectralRNN [54] learns a transition matrix with singular values in $1 \pm \epsilon$. Unfortunately, the training algorithm converged only for small $\epsilon$ thereby limiting accuracy on most datasets. Increasing the number of hidden units was found to increase accuracy somewhat but at the cost of increased training time, prediction time and model size.

**Gated RNNs**: Gated architectures [20, 13, 14, 23] achieve state-of-the-art classification accuracies by adding extra parameters but also increase model size and prediction time. This has resulted in a trend to reduce the number of gates and parameters with UGRNN [14] simplifying GRU [13] which in turn simplifies LSTM [20]. FastGRNN can be seen as a natural simplification of UGRNN where the RNN matrices are reused within the gate and are made low-rank, sparse and quantized so as to compress the model.

**Efficient training and prediction**: Efficient prediction algorithms have often been obtained by making sparsity and low-rank assumptions. Most unitary methods effectively utilize a low-rank representation of the state transition matrix to control prediction and training complexity [24, 54]. Sparsity, low-rank, and quantization were shown to be effective in RNNs [51, 39, 48], CNNs [18], trees [29] and nearest neighbour classifiers [17]. FastGRNN builds on these ideas to utilize low-rank, sparse and quantized representations for learning kilobyte sized classifiers without compromising on classification accuracy. Other approaches to speed up RNN training and prediction are based on replacing sequential hidden state transitions by parallelizable convolutions [9] or on learning skip connections [10] so as to avoid evaluating all the hidden states. Such techniques are complementary to the ones proposed in this paper and can be used to further improve FastGRNN's performance.

# 3   FastRNN and FastGRNN

**Notation**: Throughout the paper, parameters of an RNN are denoted by matrices $\mathbf{W} \in \mathbb{R}^{\hat{D} \times D}$, $\mathbf{U} \in \mathbb{R}^{\hat{D} \times \hat{D}}$ and bias vectors $\mathbf{b} \in \mathbb{R}^{\hat{D}}$, often using subscripts if multiple vectors are required to specify the architecture. $\mathbf{a} \odot \mathbf{b}$ denotes the Hadamard product between $\mathbf{a}$ and $\mathbf{b}$, i.e., $(\mathbf{a} \odot \mathbf{b})_i = \mathbf{a}_i, \mathbf{b}_i$. $\| \cdot \|_0$ denotes the number of non-zeros entries in a matrix or vector. $\| \cdot \|_F, \| \cdot \|_2$ denotes the Frobenius and spectral norm of a matrix, respectively. Unless specified, $\| \cdot \|$ denotes $\| \cdot \|_2$ of a matrix or vector. $\mathbf{a}^\top \mathbf{b} = \sum_i a_i b_i$ denotes the inner product of $\mathbf{a}$ and $\mathbf{b}$.

Standard RNN architecture [41] is known to be *unstable* for training due to exploding or vanishing gradients and hence is shunned for more expensive gated architectures.

This paper studies the FastRNN architecture that is inspired by weighted residual connections [22, 19], and shows that FastRNN can be significantly more stable and accurate than the standard RNN while preserving its prediction complexity. In particular, Section 3.1.1 demonstrates parameter settings for FastRNN that guarantee well-conditioned gradients as well as faster convergence rate and smaller generalization error than the standard RNN. This paper further strengthens FastRNN to develop the FastGRNN architecture that is more accurate than unitary methods [5, 54] and provides comparable accuracy to the state-of-the-art gated RNNs at 35x less computational cost (see Table 3).

### 3.1 FastRNN

Let $\mathbf{X} = [\mathbf{x}_1, \ldots, \mathbf{x}_T]$ be the input data where $\mathbf{x}_t \in \mathbb{R}^D$ denotes the $t$-th step feature vector. Then, the goal of multi-class RNNs is to learn a function $F : \mathbb{R}^{D \times T} \to \{1, \ldots, L\}$ that predicts one of $L$ classes for the given data point $\mathbf{X}$. Standard RNN architecture has a provision to produce an output at every time step, but we focus on the setting where each data point is associated with a single label that is predicted at the end of the time horizon $T$. Standard RNN maintains a vector of hidden state $\mathbf{h}_t \in \mathbb{R}^{\hat{D}}$ which captures temporal dynamics in the input data, i.e.,

$$\mathbf{h}_t = \tanh(\mathbf{W}\mathbf{x}_t + \mathbf{U}\mathbf{h}_{t-1} + \mathbf{b}). \tag{1}$$

As explained in the next section, learning $\mathbf{U}, \mathbf{W}$ in the above architecture is difficult as the gradient can have exponentially large (in $T$) condition number. Unitary methods explicitly control the condition number of the gradient but their training time can be significantly larger or the generated model can be less accurate.

Instead, FastRNN uses a simple weighted residual connection to stabilize the training by generating well-conditioned gradients. In particular, FastRNN updates the hidden state $\mathbf{h}_t$ as follows:

$$\tilde{\mathbf{h}}_t = \sigma(\mathbf{W}\mathbf{x}_t + \mathbf{U}\mathbf{h}_{t-1} + \mathbf{b}),$$
$$\mathbf{h}_t = \alpha\tilde{\mathbf{h}}_t + \beta\mathbf{h}_{t-1}, \tag{2}$$

where $0 \leq \alpha, \beta \leq 1$ are trainable weights that are parameterized by the sigmoid function. $\sigma : \mathbb{R} \to \mathbb{R}$ is a non-linear function such as $\tanh$, sigmoid, or ReLU, and can vary across datasets. Given $\mathbf{h}_T$, the label for a given point $\mathbf{X}$ is predicted by applying a standard classifier, e.g., logistic regression to $\mathbf{h}_T$.

Typically, $\alpha \ll 1$ and $\beta \approx 1 - \alpha$, especially for problems with larger $T$. FastRNN updates hidden state in a controlled manner with $\alpha, \beta$ limiting the extent to which the current feature vector $\mathbf{x}_t$ updates the hidden state. Also, FastRNN has only 2 more parameters than RNN and require only $\hat{D}$ more computations, which is a tiny fraction of per-step computation complexity of RNN. Unlike unitary methods [5, 23, 54], FastRNN does not introduce expensive structural constraints on $\mathbf{U}$ and hence scales well to large datasets with standard optimization techniques [28].

#### 3.1.1 Analysis

This section shows how FastRNN addresses the issue of ill-conditioned gradients, leading to stable training and smaller generalization error. For simplicity, assume that the label decision function is one dimensional and is given by $f(\mathbf{X}) = \mathbf{v}^\top \mathbf{h}_T$. Let $L(\mathbf{X}, y; \theta) = L(f(\mathbf{X}), y; \theta)$ be the logistic loss function for the given labeled data point $(\mathbf{X}, y)$ and with parameters $\theta = (\mathbf{W}, \mathbf{U}, \mathbf{v})$. Then, the gradient of $L$ w.r.t. $\mathbf{W}, \mathbf{U}, \mathbf{v}$ is given by:

$$\frac{\partial L}{\partial \mathbf{U}} = \alpha \sum_{t=0}^{T} \mathbf{D}_t \left( \prod_{k=t}^{T-1} (\alpha\mathbf{U}^\top\mathbf{D}_{k+1} + \beta\mathbf{I}) \right) (\nabla_{\mathbf{h}_T} L)\mathbf{h}_{t-1}^\top, \tag{3}$$

$$\frac{\partial L}{\partial \mathbf{W}} = \alpha \sum_{t=0}^{T} \mathbf{D}_t \left( \prod_{k=t}^{T-1} (\alpha\mathbf{U}^\top\mathbf{D}_{k+1} + \beta\mathbf{I}) \right) (\nabla_{\mathbf{h}_T} L)\mathbf{x}_t^\top, \quad \frac{\partial L}{\partial \mathbf{v}} = \frac{-y\exp{(-y \cdot \mathbf{v}^\top\mathbf{h}_T)}}{1 + \exp{(-y \cdot \mathbf{v}^\top\mathbf{h}_T)}}\mathbf{h}_T, \tag{4}$$

where $\nabla_{\mathbf{h}_T} L = -c(\theta) \cdot y \cdot \mathbf{v}$, and $c(\theta) = \frac{1}{1 + \exp{(y \cdot \mathbf{v}^\top\mathbf{h}_T)}}$. A critical term in the above expression is: $M(\mathbf{U}) = \prod_{k=t}^{T-1}(\alpha\mathbf{U}^\top\mathbf{D}_{k+1} + \beta\mathbf{I})$, whose condition number, $\kappa_{M(\mathbf{U})}$, is bounded by:

$$\kappa_{M(\mathbf{U})} \leq \frac{(1 + \frac{\alpha}{\beta}\max_k \|\mathbf{U}^\top\mathbf{D}_{k+1}\|)^{T-t}}{(1 - \frac{\alpha}{\beta}\max_k \|\mathbf{U}^\top\mathbf{D}_{k+1}\|)^{T-t}}, \tag{5}$$

where $\mathbf{D}_k = \mathrm{diag}(\sigma'(\mathbf{W}\mathbf{x}_k + \mathbf{U}\mathbf{h}_{k-1} + \mathbf{b}))$ is the Jacobian matrix of the pointwise nonlinearity. Also if $\alpha = 1$ and $\beta = 0$, which corresponds to standard RNN, the condition number of $M(\mathbf{U})$ can be as large as $(\max_k \frac{\|\mathbf{U}^\top\mathbf{D}_{k+1}\|}{\lambda_{min}(\mathbf{U}^\top\mathbf{D}_{k+1})})^{T-t}$ where $\lambda_{min}(\mathbf{A})$ denotes the minimum singular value of $\mathbf{A}$. Hence, gradient's condition number for the standard RNN can be exponential in $T$. This implies that, relative to the average eigenvalue, the gradient can explode or vanish in certain directions, leading to unstable training.

In contrast to the standard RNN, if $\beta \approx 1$ and $\alpha \approx 0$, then the condition number, $\kappa_{M(\mathbf{U})}$, for FastRNN is bounded by a small term. For example, if $\beta = 1 - \alpha$ and $\alpha = \frac{1}{T\max_k \|\mathbf{U}^\top\mathbf{D}_{k+1}\|}$,

then $\kappa_{M(\mathbf{U})} = O(1)$. Existing unitary methods are also motivated by similar observation. But they attempt to control the $\kappa_{M(\mathbf{U})}$ by restricting the condition number, $\kappa_{\mathbf{U}}$, of $\mathbf{U}$ which can still lead to ill-conditioned gradients as $\mathbf{U}^\top \mathbf{D}_{k+1}$ might still be very small in certain directions. By using residual connections, FastRNN is able to address this issue, and hence have faster training and more accurate model than the state-of-the-art unitary RNNs.

Finally, by using the above observations and a careful perturbation analysis, we can provide the following convergence and generalization error bounds for FastRNN:

**Theorem 3.1** (Convergence Bound). *Let $[(\mathbf{X}_1, y_1), \ldots, (\mathbf{X}_n, y_n)]$ be the given labeled sequential training data. Let $L(\theta) = \frac{1}{n}\sum_i L(\mathbf{X}_i, y_i; \theta)$ be the loss function with $\theta = (\mathbf{W}, \mathbf{U}, \mathbf{v})$ be the parameters of FastRNN architecture (2) with $\beta = 1 - \alpha$ and $\alpha$ such that,*

$$\alpha \leq \min\left(\frac{1}{4T \cdot |\mathcal{D}\|\mathbf{U}\|_2 - 1|}, \frac{1}{4T \cdot R_{\mathbf{U}}}, \frac{1}{T \cdot |\|\mathbf{U}\|_2 - 1|}\right),$$

*where $\mathcal{D} = \sup_{\theta,k} \|\mathbf{D}_k^\theta\|_2$. Then, randomized stochastic gradient descent [15], a minor variation of SGD, when applied to the data for a maximum of $M$ iteration outputs a solution $\widehat{\theta}$ such that:*

$$\mathbb{E}[\|\nabla_\theta L(\widehat{\theta})\|_2^2\|] \leq \mathcal{B}_M := \frac{\mathcal{O}(\alpha T) L(\theta_0)}{M} + \left(\bar{D} + \frac{4R_{\mathbf{W}} R_{\mathbf{U}} R_{\mathbf{v}}}{\bar{D}}\right)\frac{\mathcal{O}(\alpha T)}{\sqrt{M}} \leq \epsilon,$$

*where $R_{\mathbf{X}} = \max_{\mathbf{X}} \|\mathbf{X}\|_F$ for $\mathbf{X} = \{\mathbf{U}, \mathbf{W}, \mathbf{v}\}$, $L(\theta_0)$ is the loss of the initial classifier, and the step-size of the $k$-th SGD iteration is fixed as: $\gamma_k = \min\left\{\frac{1}{\mathcal{O}(\alpha T)}, \frac{\bar{D}}{T\sqrt{M}}\right\}, k \in [M], \bar{D} \geq 0$. Maximum number of iterations is bounded by $M = O(\frac{\alpha T}{\epsilon^2} \cdot poly(L(\theta_0), R_{\mathbf{W}} R_{\mathbf{U}} R_{\mathbf{v}}, \bar{D})), \epsilon \geq 0$.*

**Theorem 3.2** (Generalization Error Bound). *[6] Let $\mathcal{Y}, \hat{\mathcal{Y}} \subseteq [0, 1]$ and let $\mathcal{F}_T$ denote the class of FastRNN with $\|\mathbf{U}\|_F \leq R_{\mathbf{U}}, \|\mathbf{W}\|_F \leq R_{\mathbf{W}}$. Let the final classifier be given by $\sigma(\mathbf{v}^\top \mathbf{h}_T)$, $\|\mathbf{v}\|_2 \leq R_{\mathbf{v}}$. Let $L : \mathcal{Y} \times \hat{\mathcal{Y}} \rightarrow [0, B]$ be any 1-Lipschitz loss function. Let $D$ be any distribution on $\mathcal{X} \times \mathcal{Y}$ such that $\|\mathbf{x}_{it}\|_2 \leq R_x$ a.s. Let $0 \leq \delta \leq 1$. For all $\beta = 1 - \alpha$ and $\alpha$ such that,*

$$\alpha \leq \min\left(\frac{1}{4T \cdot |\mathcal{D}\|\mathbf{U}\|_2 - 1|}, \frac{1}{4T \cdot R_{\mathbf{U}}}, \frac{1}{T \cdot |\|\mathbf{U}\|_2 - 1|}\right),$$

*where $\mathcal{D} = \sup_{\theta,k} \|\mathbf{D}_k^\theta\|_2$, we have that with probability at least $1 - \delta$, all functions $f \in \mathbf{v} \circ \mathcal{F}_T$ satisfy,*

$$\mathbb{E}_D[L(f(\mathbf{X}), y)] \leq \frac{1}{n}\sum_{i=1}^n L(f(\mathbf{X}_i), y_i) + \mathcal{C}\frac{O(\alpha T)}{\sqrt{n}} + B\sqrt{\frac{\ln(\frac{1}{\delta})}{n}},$$

*where $\mathcal{C} = R_{\mathbf{W}} R_{\mathbf{U}} R_{\mathbf{x}} R_{\mathbf{v}}$ represents the boundedness of the parameter matrices and the data.*

The convergence bound states that if $\alpha = O(1/T)$ then the algorithm converges to a stationary point in *constant time* with respect to $T$ and polynomial time with respect to all the other problem parameters. Generalization bound states that for $\alpha = O(1/T)$, the generalization error of FastRNN is *independent* of $T$. In contrast, *similar proof technique* provide exponentially poor (in $T$) error bound and convergence rate for standard RNN. But, this is an upper bound, so potentially significantly better error bounds for RNN might exist; matching lower bound results for standard RNN is an interesting research direction. Also, $O(T^2)$ generalization error bound can be argued using VC-dimension style arguments [4]. But such bounds hold for specific settings like binary $y$, and are independent of problem hardness parameterized by the size of the weight matrices ($R_{\mathbf{W}}, R_{\mathbf{U}}$).

Finally, note that the above analysis fixes $\alpha = O(1/T), \beta = 1 - \alpha$, but in practice FastRNN learns $\alpha, \beta$ (which is similar to performing cross-validation on $\alpha, \beta$). However, interestingly, across datasets the learnt $\alpha, \beta$ values indeed display a similar scaling wrt $T$ for large $T$ (see Figure 2).

### 3.2 FastGRNN

While FastRNN controls the condition number of gradient reasonably well, its expressive power might be limited for some datasets. This concern is addressed by a novel architecture, FastGRNN, that uses a scalar weighted residual connection for each and every coordinate of the hidden state $\mathbf{h}_t$. That is,

$$\mathbf{z}_t = \sigma(\mathbf{W}\mathbf{x}_t + \mathbf{U}\mathbf{h}_{t-1} + \mathbf{b}_z),$$
$$\tilde{\mathbf{h}}_t = \tanh(\mathbf{W}\mathbf{x}_t + \mathbf{U}\mathbf{h}_{t-1} + \mathbf{b}_h),$$
$$\mathbf{h}_t = (\zeta(\mathbf{1} - \mathbf{z}_t) + \nu) \odot \tilde{\mathbf{h}}_t + \mathbf{z}_t \odot \mathbf{h}_{t-1}, \tag{6}$$

where $0 \leq \zeta, \nu \leq 1$ are trainable parameters that are parameterized by the sigmoid function, and $\sigma : \mathbb{R} \rightarrow \mathbb{R}$ is a non-linear function such as $\tanh$, sigmoid and can vary across datasets. Note that each coordinate of $\mathbf{z}_t$ is similar to parameter $\beta$ in (2) and $\zeta(\mathbf{1} - \mathbf{z}_t) + \nu$'s coordinates simulate $\alpha$ parameter; also if $\nu \approx 0, \zeta \approx 1$ then it satisfies the intuition that $\alpha + \beta = 1$. It was observed that across all datasets, this gating mechanism outperformed the simple vector extension of FastRNN where each coordinate of $\alpha$ and $\beta$ is learnt (see Appendix G).

FastGRNN computes each coordinate of gate $\mathbf{z}_t$ using a non-linear function of $\mathbf{x}_t$ and $\mathbf{h}_{t-1}$. To minimize the number of parameters, FastGRNN reuses the matrices $\mathbf{W}, \mathbf{U}$ for the vector-valued gating function as well. Hence, FastGRNN's inference complexity is almost same as that of the standard RNN but its accuracy and training stability is on par with expensive gated architectures like GRU and LSTM.

**Sparse low-rank representation**: FastGRNN further compresses the model size by using a low-rank and a sparse representation of the parameter matrices $\mathbf{W}, \mathbf{U}$. That is,

$$\mathbf{W} = \mathbf{W}^1(\mathbf{W}^2)^\top, \ \mathbf{U} = \mathbf{U}^1(\mathbf{U}^2)^\top, \ \|\mathbf{W}^i\|_0 \leq s_w^i, \ \|\mathbf{U}^i\|_0 \leq s_u^i, \ i = \{1, 2\}, \qquad (7)$$

where $\mathbf{W}^1 \in \mathbb{R}^{\hat{D} \times r_w}, \mathbf{W}^2 \in \mathbb{R}^{D \times r_w}$, and $\mathbf{U}^1, \mathbf{U}^2 \in \mathbb{R}^{\hat{D} \times r_u}$. Hyperparameters $r_w, s_w, r_u, s_u$ provide an efficient way to control the *accuracy-memory* trade-off for FastGRNN and are typically set via fine-grained validation. In particular, such compression is critical for FastGRNN model to fit on resource-constrained devices. Second, this low-rank representation brings down the prediction time by reducing the cost at each time step from $\mathcal{O}(\hat{D}(D + \hat{D}))$ to $\mathcal{O}(r_w(D + \hat{D}) + r_u\hat{D})$. This enables FastGRNN to provide on-device prediction in real-time on battery constrained devices.

### 3.2.1 Training FastGRNN

The parameters for FastGRNN: $\mathbf{\Theta}_{\text{FastGRNN}} = (\mathbf{W}^i, \mathbf{U}^i, \mathbf{b}_h, \mathbf{b}_z, \zeta, \nu)$ are trained jointly using projected batch stochastic gradient descent (b-SGD) (or other stochastic optimization methods) with typical batch sizes ranging from $64 - 128$. In particular, the optimization problem is given by:

$$\min_{\mathbf{\Theta}_{\text{FastGRNN}}, \|\mathbf{W}^i\|_0 \leq s_w^i, \|\mathbf{U}^i\|_0 \leq s_u^i, i \in \{1,2\}} \mathcal{J}(\mathbf{\Theta}_{\text{FastGRNN}}) = \frac{1}{n} \sum_j L(\mathbf{X}_j, y_j; \mathbf{\Theta}_{\text{FastGRNN}}) \qquad (8)$$

where $L$ denotes the appropriate loss function (typically softmax cross-entropy). The training procedure for FastGRNN is divided into 3 stages:

**(I) Learning low-rank representation (L)**: In the first stage of the training, FastGRNN is trained for $e_1$ epochs with the model as specified by (7) using b-SGD. This stage of optimization ignores the sparsity constraints on the parameters and learns a low-rank representation of the parameters.

**(II) Learning sparsity structure (S)**: FastGRNN is next trained for $e_2$ epochs using b-SGD, projecting the parameters onto the space of sparse low-rank matrices after every few batches while maintaining support between two consecutive projection steps. This stage, using b-SGD with Iterative Hard Thresholding (IHT), helps FastGRNN identify the correct support for parameters $(\mathbf{W}^i, \mathbf{U}^i)$.

**(III) Optimizing with fixed parameter support**: In the last stage, FastGRNN is trained for $e_3$ epochs with b-SGD while freezing the support set of the parameters.

In practice, it is observed that $e_1 = e_2 = e_3 = 100$ generally leads to the convergence of FastGRNN to a good solution. Early stopping is often deployed in stages (II) and (III) to obtain the best models.

### 3.3 Byte Quantization (Q)

FastGRNN further compresses the model by quantizing each element of $\mathbf{W}^i, \mathbf{U}^i$, restricting them to at most one byte along with byte indexing for sparse models. However, simple integer quantization of $\mathbf{W}^i, \mathbf{U}^i$ leads to a large loss in accuracy due to gross approximation. Moreover, while such a quantization reduces the model size, the prediction time can still be large as non-linearities will require all the hidden states to be floating point. FastGRNN overcomes these shortcomings by training $\mathbf{W}^i$ and $\mathbf{U}^i$ using *piecewise-linear* approximation of the non-linear functions, thereby ensuring that all the computations can be performed with integer arithmetic. During training, FastGRNN replaces the non-linear function in (6) with their respective approximations and uses the above mentioned training procedure to obtain $\mathbf{\Theta}_{\text{FastGRNN}}$. The floating point parameters are then jointly quantized to ensure that all the relevant entities are integer-valued and the entire inference computation can

be executed efficiently with integer arithmetic without a significant drop in accuracy. For instance, Tables 4, 5 show that on several datasets FastGRNN models are 3-4x faster than their corresponding FastGRNN-Q models on common IoT boards with no floating point unit (FPU). FastGRNN-LSQ, FastGRNN "minus" the Low-rank, Sparse and Quantized components, is the base model with no compression.

## 4 Experiments

**Datasets**: FastRNN and FastGRNN's performance was benchmarked on the following IoT tasks where having low model sizes and prediction times was critical to the success of the application: (a) Wakeword-2 [45] - detecting utterances of the "Hey Cortana" wakeword; (b) Google-30 [49] and Google-12 - detection of utterances of 30 and 10 commands plus background noise and silence and (c) HAR-2 [3] and DSA-19 [2] - Human Activity Recognition (HAR) from an accelerometer and gyroscope on a Samsung Galaxy S3 smartphone and Daily and Sports Activity (DSA) detection from a resource-constrained IoT wearable device with 5 Xsens MTx sensors having accelerometers, gyroscopes and magnetometers on the torso and four limbs. Traditional RNN tasks typically do not have prediction constraints and are therefore not the focus of this paper. Nevertheless, for the sake of completeness, experiments were also carried out on benchmark RNN tasks such as language modeling on the Penn Treebank (PTB) dataset [33], star rating prediction on a scale of 1 to 5 of Yelp reviews [52] and classification of MNIST images on a pixel-by-pixel sequence [32, 31].

All datasets, apart from Wakeword-2, are publicly available and their pre-processing and feature extraction details are provided in Appendix B. The publicly provided training set for each dataset was subdivided into $80\%$ for training and $20\%$ for validation. Once the hyperparameters had been fixed, the algorithms were trained on the full training set and results were reported on the publicly available test set. Table 1 lists the statistics of all datasets.

**Baseline algorithms and Implementation**: FastRNN and FastGRNN were compared to standard RNN [41], leading unitary RNN approaches such as SpectralRNN [54], Orthogonal RNN (oRNN) [37], Efficient Unitary Recurrent Neural Networks (EURNN) [24], FactoredRNN [47] and state-of-the-art gated RNNs including UGRNN [14], GRU [13] and LSTM [20]. Details of these methods are provided in Section 2. Native Tensorflow implementations were used for the LSTM and GRU architectures. For all the other RNNs, publicly available implementations provided by the authors were used taking care to ensure that published results could be reproduced thereby verifying the code and hyper-parameter settings. All experiments were run on an Nvidia Tesla P40 GPU with CUDA 9.0 and cuDNN 7.1 on a machine with an Intel Xeon 2.60 GHz CPU with 12 cores.

**Hyper-parameters**: The hyper-parameters of each algorithm were set by a fine-grained validation wherever possible or according to the settings recommended by the authors otherwise. Adam, Nesterov Momentum and SGD were used to optimize each algorithm on each dataset and the optimizer with the best validation performance was selected. The learning rate was initialized to $10^{-2}$ for all architectures except for RNNs where the learning rate was initialized to $10^{-3}$ to ensure stable training. Each algorithm was run for 200 epochs after which the learning rate was decreased by a factor of $10^{-1}$ and the algorithm run again for another 100 epochs. This procedure was carried out on all datasets except for Pixel MNIST where the learning rate was decayed by $\frac{1}{2}$ after each pass of 200 epochs. Batch sizes between 64 and 128 training points were tried for most architectures and a batch size of 100 was found to work well in general except for standard RNNs which required a batch size of 512. FastRNN used $\tanh$ as the non-linearity in most cases except for a few (indicated by $^+$) where ReLU gave slightly better results. Table 11 in the Appendix lists the non-linearity, optimizer and hyper-parameter settings for FastGRNN on all datasets.

<table>
<tr><td colspan="5">Table 1: Dataset Statistics</td></tr>
<tr><td>Dataset</td><td>#Train</td><td>#Features</td><td>#Time Steps</td><td>#Test</td></tr>
<tr><td>Google-12</td><td>22,246</td><td>3,168</td><td>99</td><td>3,081</td></tr>
<tr><td>Google-30</td><td>51,088</td><td>3,168</td><td>99</td><td>6,835</td></tr>
<tr><td>Wakeword-2</td><td>195,800</td><td>5,184</td><td>162</td><td>83,915</td></tr>
<tr><td>Yelp-5</td><td>500,000</td><td>38,400</td><td>300</td><td>500,000</td></tr>
<tr><td>HAR-2</td><td>7,352</td><td>1,152</td><td>128</td><td>2,947</td></tr>
<tr><td>Pixel-MNIST-10</td><td>60,000</td><td>784</td><td>784</td><td>10,000</td></tr>
<tr><td>PTB-10000</td><td>929,589</td><td>—</td><td>300</td><td>82,430</td></tr>
<tr><td>DSA-19</td><td>4,560</td><td>5,625</td><td>125</td><td>4,560</td></tr>
</table>

<table>
<tr><td colspan="5">Table 2: PTB Language Modeling - 1 Layer</td></tr>
<tr><td>Method</td><td>Test Perplexity</td><td>Train Perplexity</td><td>Model Size (KB)</td><td>Train Time (min)</td></tr>
<tr><td>RNN</td><td>144.71</td><td>68.11</td><td>129</td><td>**9.11**</td></tr>
<tr><td>FastRNN</td><td>127.76$^+$</td><td>109.07</td><td>513</td><td>11.20</td></tr>
<tr><td>FastGRNN-LSQ</td><td>**115.92**</td><td>89.58</td><td>513</td><td>12.53</td></tr>
<tr><td>FastGRNN</td><td>116.11</td><td>81.31</td><td>**39**</td><td>13.75</td></tr>
<tr><td>SpectralRNN</td><td>130.20</td><td>65.42</td><td>242</td><td>—</td></tr>
<tr><td>UGRNN</td><td>119.71</td><td>65.25</td><td>256</td><td>11.12</td></tr>
<tr><td>LSTM</td><td>117.41</td><td>69.44</td><td>2052</td><td>13.52</td></tr>
</table>

**Evaluation criteria**: The emphasis in this paper is on designing RNN architectures which can run on low-memory IoT devices and which are efficient at prediction time. As such, the model size of each architecture is reported along with its training time and classification accuracy (F1 score on the Wakeword-2 dataset and perplexity on the PTB dataset). Prediction times on some of the popular IoT boards are also reported. Note that, for NLP applications such as PTB and Yelp, just the model size of the various RNN architectures has been reported. In a real application, the size of the learnt word-vector embeddings (10 MB for FastRNN and FastGRNN) would also have to be considered.

**Results**: Tables 2 and 3 compare the performance of FastRNN, FastGRNN and FastGRNN-LSQ to state-of-the-art RNNs. Three points are worth noting about FastRNN's performance. First, FastRNN's prediction accuracy gains over a standard RNN ranged from 2.34% on the Pixel-MNIST dataset to 19% on the Google-12 dataset. Second, FastRNN's prediction accuracy could surpass leading unitary RNNs on 6 out of the 8 datasets with gains up to 2.87% and 3.77% over SpectralRNN on the Google-12 and DSA-19 datasets respectively. Third, FastRNN's training speedups over all unitary and gated RNNs could range from 1.2x over UGRNN on the Yelp-5 and DSA-19 datasets to 196x over EURNN on the Google-12 dataset. This demonstrates that the vanishing and exploding gradient problem could be overcome by the addition of a simple weighted residual connection to the standard RNN architecture thereby allowing FastRNN to train efficiently and stably. This also demonstrates that the residual connection offers a theoretically principled architecture that can often result in accuracy gains without limiting the expressive power of the hidden state transition matrix.

Tables 2 and 3 also demonstrate that FastGRNN-LSQ could be more accurate and faster to train than all unitary RNNs. Furthermore, FastGRNN-LSQ could match the accuracies and training times of state-of-the-art gated RNNs while having models that could be 1.18-4.87x smaller. This demonstrates that extending the residual connection to a gate which reuses the RNN matrices increased accuracy with virtually no increase in model size over FastRNN in most cases. In fact, on Google-30 and Pixel-MNIST FastGRNN-LSQ's model size was lower than FastRNN's as it had a lower hidden dimension indicating that the gate efficiently increased expressive power.

Finally, Tables 2 and 3 show that FastGRNN's accuracy was at most 1.13% worse than the best RNN but its model could be up to 35x smaller even as compared to low-rank unitary methods such as SpectralRNN. Figures 3 and 4 in the Appendix also show that FastGRNN-LSQ and FastGRNN's classification accuracies could be higher than those obtained by the best unitary and gated RNNs for any given model size in the 0-128 KB range. This demonstrates the effectiveness of making FastGRNN's parameters low-rank, sparse and quantized and allows FastGRNN to fit on the Arduino

Table 3: FastGRNN had up to 35x smaller models than leading RNNs with almost no loss in accuracy

| | Dataset | Google-12 | | | Google-30 | | | Wakeword-2 | | |
|---|---|---|---|---|---|---|---|---|---|---|
| | Method | Accuracy (%) | Model Size (KB) | Train Time (hr) | Accuracy (%) | Model Size (KB) | Train Time (hr) | F1 Score | Model Size (KB) | Train Time(hr) |
| | RNN | 73.25 | 56 | 1.11 | 80.05 | 63 | 2.13 | 89.17 | 8 | **0.28** |
| Proposed | FastRNN | 92.21[+] | 56 | **0.61** | 91.60[+] | 96 | **1.30** | 97.09 | 8 | 0.69 |
| | FastGRNN-LSQ | **93.18** | 57 | 0.63 | **92.03** | 45 | 1.41 | **98.19** | 8 | 0.83 |
| | FastGRNN | 92.10 | **5.5** | 0.75 | 90.78 | **6.25** | 1.77 | 97.83 | **1** | 1.08 |
| Unitary | SpectralRNN | 91.59 | 228 | 19.00 | 88.73 | 128 | 11.00 | 96.75 | 17 | 7.00 |
| | EURNN | 76.79 | 210 | 120.00 | 56.35 | 135 | 19.00 | 92.22 | 24 | 69.00 |
| | oRNN | 88.18 | 102 | 16.00 | 86.95 | 120 | 35.00 | — | — | — |
| | FactoredRNN | 53.33 | 1114 | 7.00 | 40.57 | 1150 | 8.52 | — | — | — |
| Gated | UGRNN | 92.63 | 75 | 0.78 | 90.54 | 260 | 2.11 | 98.17 | 16 | 1.00 |
| | GRU | 93.15 | 248 | 1.23 | 91.41 | 257 | 2.70 | 97.63 | 24 | 1.38 |
| | LSTM | 92.30 | 212 | 1.36 | 90.31 | 219 | 2.63 | 97.82 | 32 | 1.71 |

| | Dataset | Yelp-5 | | | HAR-2 | | | DSA-19 | | | Pixel-MNIST-10 | | |
|---|---|---|---|---|---|---|---|---|---|---|---|---|---|
| | Method | Accuracy (%) | RNN Model Size (KB) | Train Time (hr) | Accuracy (%) | Model Size (KB) | Train Time (hr) | Accuracy (%) | Model Size (KB) | Train Time (min) | Accuracy (%) | Model Size (KB) | Train Time (hr) |
| | RNN | 47.59 | 130 | **3.33** | 91.31 | 29 | 0.11 | 71.68 | 20 | **1.11** | 94.10 | 71 | 45.56 |
| Proposed | FastRNN | 55.38 | 130 | 3.61 | 94.50[+] | 29 | **0.06** | 84.14 | 97 | 1.92 | 96.44 | 166 | 15.10 |
| | FastGRNN-LSQ | **59.51** | 130 | 3.91 | 95.38 | 29 | 0.08 | **85.00** | 208 | 2.15 | **98.72** | 71 | 12.57 |
| | FastGRNN | 59.43 | **8** | 4.62 | **95.59** | 3 | 0.10 | 83.73 | **3.25** | 2.10 | 98.20 | **6** | 16.97 |
| Unitary | SpectralRNN | 56.56 | 89 | 4.92 | 95.48 | 525 | 0.73 | 80.37 | 50 | 2.25 | 97.70 | 25 | — |
| | EURNN | 59.01 | 122 | 72.00 | 93.11 | 12 | 0.84 | — | — | — | 95.38 | 64 | 122.00 |
| | oRNN | — | — | — | 94.57 | 22 | 2.72 | 72.52 | 18 | — | 97.20 | 49 | — |
| | FactoredRNN | — | — | — | 78.65 | **1** | 0.11 | 73.20 | 1154 | — | 94.60 | 125 | — |
| Gated | UGRNN | 58.67 | 258 | 4.34 | 94.53 | 37 | 0.12 | 84.74 | 399 | 2.31 | 97.29 | 84 | 15.17 |
| | GRU | 59.02 | 388 | 8.12 | 93.62 | 71 | 0.13 | 84.84 | 270 | 2.33 | 98.70 | 123 | 23.67 |
| | LSTM | 59.49 | 516 | 8.61 | 93.65 | 74 | 0.18 | 84.84 | 526 | 2.58 | 97.80 | 265 | 26.57 |

| Table 4: Prediction time in ms on the Arduino MKR1000 |||| Table 5: Prediction time in ms on the Arduino Due ||||
| Method | Google-12 | HAR-2 | Wakeword-2 | Method | Google-12 | HAR-2 | Wakeword-2 |
|---|---|---|---|---|---|---|---|
| FastGRNN | 537 | 162 | 175 | FastGRNN | 242 | 62 | 77 |
| FastGRNN-Q | 2282 | 553 | 755 | FastGRNN-Q | 779 | 172 | 238 |
| RNN | 12028 | 2249 | 2232 | RNN | 3472 | 590 | 653 |
| UGRNN | 22875 | 4207 | 6724 | UGRNN | 6693 | 1142 | 1823 |
| SpectralRNN | 70902 | — | 10144 | SpectralRNN | 17766 | 55558 | 2691 |

Uno having just 2 KB RAM and 32 KB flash memory. In particular, FastGRNN was able to recognize the "Hey Cortana" wakeword just as accurately as leading RNNs but with a 1 KB model.

**Prediction on IoT boards**: Unfortunately, most RNNs were too large to fit on an Arduino Uno apart from FastGRNN. On the slightly more powerful Arduino MKR1000 having an ARM Cortex M0+ microcontroller operating at 48 MHz with 32 KB RAM and 256 KB flash memory, Table 4 shows that FastGRNN could achieve the same prediction accuracy while being 25-45x faster at prediction than UGRNN and 57-132x faster than SpectralRNN. Results on the even more powerful Arduino Due are presented in Table 5 while results on the Raspberry Pi are presented in Table 12 of the Appendix.

**Ablations, extensions and parameter settings**: Enforcing that FastGRNN's matrices be low-rank led to a slight increase in prediction accuracy and reduction in prediction costs as shown in the ablation experiments in Tables 8, 9 and 10 in the Appendix. Adding sparsity and quantization led to a slight drop in accuracy but resulted in significantly smaller models. Next, Table 16 in the Appendix shows that regularization and layering techniques [36] that have been proposed to increase the prediction accuracy of other gated RNNs are also effective for FastGRNN and can lead to reductions in perplexity on the PTB dataset. Finally, Figure 2 and Table 7 of the Appendix measure the agreement between FastRNN's theoretical analysis and empirical observations. Figure 2 (a) shows that the $\alpha$ learnt on datasets with $T$ time steps is decreasing function of $T$ and Figure 2 (b) shows that the learnt $\alpha$ and $\beta$ follow the relation $\alpha/\beta \approx O(1/T)$ for large $T$ which is one of the settings in which FastRNN's gradients stabilize and training converges quickly as proved by Theorems 3.1 and 3.2. Furthermore, $\beta$ can be seen to be close to $1 - \alpha$ for large $T$ in Figure 2 (c) as assumed in Section 3.1.1 for the convergence of long sequences. For instance, the relative error between $\beta$ and $1 - \alpha$ for Google-12 with 99 timesteps was 2.15%, for HAR-2 with 128 timesteps was 3.21% and for MNIST-10 with 112 timesteps was 0.68%. However, for short sequences where there was a lower likelihood of gradients exploding or vanishing, $\beta$ was found to deviate significantly from $1 - \alpha$ as this led to improved prediction accuracy. Enforcing that $\beta = 1 - \alpha$ on short sequences was found to drop accuracy by up to 1.5%.

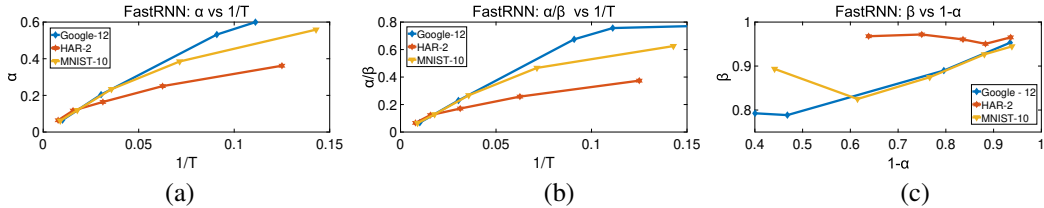

Figure 2: Plots (a) and (b) show the variation of $\alpha$ and $\alpha/\beta$ of FastRNN with respect to $1/T$ for three datasets. Plot (c) shows the relation between $\beta$ and $1 - \alpha$. In accordance with Theorem 3.1, the learnt values of $\alpha$ and $\alpha/\beta$ scale as $O(1/T)$ while $\beta \to 1 - \alpha$ for long sequences.

## 5 Conclusions

This paper proposed the FastRNN and FastGRNN architectures for efficient RNN training and prediction. FastRNN could lead to provably stable training by incorporating a residual connection with two scalar parameters into the standard RNN architecture. FastRNN was demonstrated to have lower training times, lower prediction costs and higher prediction accuracies than leading unitary RNNs in most cases. FastGRNN extended the residual connection to a gate reusing the RNN matrices and was able to match the accuracies of state-of-the-art gated RNNs but with significantly lower prediction costs. FastGRNN's model could be compressed to 1-6 KB without compromising accuracy in many cases by enforcing that its parameters be low-rank, sparse and quantized. This allowed FastGRNN to make accurate predictions efficiently on severely resource-constrained IoT devices too tiny to hold other RNN models.

## Acknowledgements

We are grateful to Ankit Anand, Niladri Chatterji, Kunal Dahiya, Don Dennis, Inderjit S. Dhillon, Dinesh Khandelwal, Shishir Patil, Adithya Pratapa, Harsha Vardhan Simhadri and Raghav Somani for helpful discussions and feedback. KB acknowledges the support of the NSF through grant IIS-1619362 and of the AFOSR through grant FA9550-17-1-0308.

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
