[Supplementary Material · FastGRNN-supplementary.pdf]

# A Convergence Analysis for FastRNN

---

**Algorithm 1:** Randomized Stochastic Gradient

---

**Input:** Initial point $\theta_1$, iteration limit M, step sizes $\gamma_{k \geq 1}$, Probability mass function $P_R(\cdot)$ supported on $\{1, 2, \ldots, M\}$
**Initialize:** $R$ be a random variable with probability mass function $P_R$
**for** $m = 1, \ldots, R$ **do**
     Obtain sample of stochastic gradient $\nabla L_t(\theta_t)$
     $\theta_t \leftarrow \theta_{t-1} - \gamma_t \nabla L_t(\theta_t)$
**Output:** $\theta_R$

---

Let $\theta = (\mathbf{W}, \mathbf{U}, \mathbf{v})$ represent the set of parameters of the scalar gated recurrent neural network. In order to prove the convergence properties of Randomized Stochastic Gradient (see Algorithm 1) as in [15], we first obtain a bound on the Lipschitz constant of the loss function $L(\mathbf{X}, y; \theta) := \log(1 + \exp(-y \cdot \mathbf{v}^\top \mathbf{h}_T))$ where $\mathbf{h}_T$ is the output of FastRNN after $T$ time steps given input $\mathbf{X}$.

The gradient $\nabla_\theta L$ of the loss function is given by $(\frac{\partial L}{\partial \mathbf{W}}, \frac{\partial L}{\partial \mathbf{U}}, \frac{\partial L}{\partial \mathbf{v}})$ wherein

$$\frac{\partial L}{\partial \mathbf{U}} = \alpha \sum_{t=0}^{T} \mathbf{D}_t \left( \prod_{k=t}^{T-1} (\alpha \mathbf{U}^\top \mathbf{D}_{k+1} + \beta \mathbf{I}) \right) (\nabla_{\mathbf{h}_T} L) \mathbf{h}_{t-1}^\top \tag{9}$$

$$\frac{\partial L}{\partial \mathbf{W}} = \alpha \sum_{t=0}^{T} \mathbf{D}_t \left( \prod_{k=t}^{T-1} (\alpha \mathbf{U}^\top \mathbf{D}_{k+1} + \beta \mathbf{I}) \right) (\nabla_{\mathbf{h}_T} L) \mathbf{x}_t^\top \tag{10}$$

$$\frac{\partial L}{\partial \mathbf{v}} = \frac{-y \exp(-y \cdot \mathbf{v}^\top \mathbf{h}_T)}{1 + \exp(-y \cdot \mathbf{v}^\top \mathbf{h}_T)} \mathbf{h}_T, \tag{11}$$

where $\nabla_{\mathbf{h}_T} L = -c(\theta) y \cdot \mathbf{v}$, with $c(\theta) = \frac{1}{1+\exp(y \cdot \mathbf{v}^\top \mathbf{h}_T)}$. We do a perturbation analysis and obtain a bound on $\|\nabla_\theta L(\theta) - \nabla_\theta L(\theta + \delta)\|_2$ where $\delta = (\delta_{\mathbf{W}}, \delta_{\mathbf{U}}, \delta_{\mathbf{v}})$.

**Deviation bound for $\mathbf{h}_T$:** In this subsection, we consider bounding the term $\|\mathbf{h}_T(\theta + \delta) - \mathbf{h}_T(\theta)\|_2$ evaluated on the same input $\mathbf{X}$. Note that for FastRNN, $\mathbf{h}_T = \alpha \tilde{\mathbf{h}}_T + \beta \mathbf{h}_{T-1}$. For notational convenience, we use $\mathbf{h}_T' = \mathbf{h}_T(\theta + \delta)$ and $\mathbf{h}_T = \mathbf{h}_T(\theta)$.

$$\|\mathbf{h}_T' - \mathbf{h}_T\|_2 \leq \beta \|\mathbf{h}_{T-1}' - \mathbf{h}_{T-1}\|_2 + \alpha \|\sigma(\mathbf{W}\mathbf{x}_T + \mathbf{U}\mathbf{h}_{T-1}) - \sigma((\mathbf{W} + \delta_{\mathbf{W}})\mathbf{x}_T + (\mathbf{U} + \delta_{\mathbf{U}})\mathbf{h}_{T-1}')\|_2$$

$$\overset{\zeta_1}{\leq} \beta \|\mathbf{h}_{T-1}' - \mathbf{h}_{T-1}\|_2 + \alpha \|\mathbf{U}\mathbf{h}_{T-1} - \delta_{\mathbf{W}}\mathbf{x}_T - \mathbf{U}\mathbf{h}_{T-1}' - \delta_{\mathbf{U}}\mathbf{h}_{T-1}'\|_2$$

$$\leq (\alpha \|\mathbf{U}\|_2 + \beta)\|\mathbf{h}_{T-1}' - \mathbf{h}_{T-1}\|_2 + \alpha(\sqrt{\hat{D}} \cdot \|\delta_{\mathbf{U}}\|_2 + \|\delta_{\mathbf{W}}\|_2 R_{\mathbf{x}})$$

$$\vdots$$

$$\leq \alpha(\sqrt{\hat{D}} \cdot \|\delta_{\mathbf{U}}\|_2 + \|\delta_{\mathbf{W}}\|_2 R_{\mathbf{x}}) \left( 1 + (\alpha\|\mathbf{U}\|_2 + \beta) + \ldots + (\alpha\|\mathbf{U}\|_2 + \beta)^{T-1} \right)$$

$$\overset{\zeta_2}{\leq} \alpha(\sqrt{\hat{D}} \cdot \|\delta_{\mathbf{U}}\|_2 + \|\delta_{\mathbf{W}}\|_2 R_{\mathbf{x}}) \frac{(\alpha(\|\mathbf{U}\|_2 - 1) + 1)^T - 1}{\alpha(\|\mathbf{U}\|_2 - 1)} \leq 2\alpha(\sqrt{\hat{D}} \cdot \|\delta_{\mathbf{U}}\|_2 + \|\delta_{\mathbf{W}}\|_2 R_{\mathbf{x}}) \cdot T$$

$$\leq \frac{2\sqrt{\hat{D}} \cdot \|\delta_{\mathbf{U}}\|_2 + 2\|\delta_{\mathbf{W}}\|_2 R_{\mathbf{x}}}{|\|\mathbf{U}\|_2 - 1|}, \tag{12}$$

where $\zeta_1$ follows by using 1-lipschitz property of the sigmoid function and $\zeta_2$ follows by setting $\alpha = O(\frac{1}{T \cdot |\|\mathbf{U}\|_2 - 1|})$ and $\beta = 1 - \alpha$.

**Deviation bound for $c(\theta)$:** In this subsection, we consider bounding the deviation $c(\theta) - c(\theta + \delta)$.

$$|c(\theta) - c(\theta + \delta)| \leq |\mathbf{v}^\top \mathbf{h}_T - (\mathbf{v} + \delta_{\mathbf{v}}^\top)\mathbf{h}_T'|$$

$$\leq |\mathbf{v}^\top(\mathbf{h}_T - \mathbf{h}_T')| + \|\delta_{\mathbf{v}}\|_2 \|\mathbf{h}_t\|_2$$

$$\leq \|\mathbf{v}\|_2 \|\mathbf{h}_T - \mathbf{h}_T'\|_2 + \|\delta_{\mathbf{v}}\|_2 \|\mathbf{h}_t\|_2$$

$$\leq R_{\mathbf{v}} \|\mathbf{h}_T - \mathbf{h}_T'\|_2 + \sqrt{\hat{D}} \|\delta_{\mathbf{v}}\|_2. \tag{13}$$

**Deviation bound for $\frac{\partial L}{\partial \mathbf{v}}$:** In this subsection we consider the bounds on $\|\frac{\partial L}{\partial \mathbf{v}}(\theta) - \frac{\partial L}{\partial \mathbf{v}}(\theta + \delta)\|_2$.

$$
\begin{aligned}
\left\| \frac{\partial L}{\partial \mathbf{v}}(\theta) - \frac{\partial L}{\partial \mathbf{v}}(\theta + \delta) \right\|_2 &= \left\| \frac{\mathbf{h}_T}{1 + \exp(y\mathbf{v}^\top \mathbf{h}_T)} - \frac{\mathbf{h}'_T}{1 + \exp(y(\mathbf{v} + \delta_\mathbf{v})^\top \mathbf{h}'_T)} \right\|_2 \\
&= \left\| c(\theta)\mathbf{h}_T - c(\theta + \delta)\mathbf{h}'_T \right\|_2 \\
&= \left\| (c(\theta) - c(\theta + \delta)) \cdot \mathbf{h}_T + c(\theta + \delta) \cdot (\mathbf{h}_T - \mathbf{h}'_T) \right\|_2 \\
&\leq \sqrt{\hat{D}} \cdot |c(\theta) - c(\theta + \delta)| + \|\mathbf{h}_T - \mathbf{h}'_T\|_2 \\
&\leq \left( \sqrt{\hat{D}} R_\mathbf{v} + 1 \right) \cdot \|\mathbf{h}_T - \mathbf{h}'_T\|_2 + \hat{D}\|\delta_\mathbf{v}\|_2. \tag{14}
\end{aligned}
$$

**Deviation bound for $\frac{\partial L}{\partial \mathbf{W}}$:** In this subsection, we analyze $\|\frac{\partial L}{\partial \mathbf{W}}(\theta) - \frac{\partial L}{\partial \mathbf{W}}(\theta + \delta)\|_2$. Let $\mathcal{D} = \sup_{k,\theta} \|\mathbf{D}_k^\theta\|$.

$$
\begin{aligned}
&\left\| \frac{\partial L}{\partial \mathbf{W}}(\theta) - \frac{\partial L}{\partial \mathbf{W}}(\theta + \delta) \right\|_F \\
&= \alpha R_\mathbf{x} \left\| \sum_{t=0}^{T} \left[ \left( c(\theta)\mathbf{D}_t^\theta \prod_{k=t}^{T-1} (\alpha \mathbf{U}^\top \mathbf{D}_{k+1}^\theta + \beta \mathbf{I}) \right) \mathbf{v} - \left( c(\theta + \delta)\mathbf{D}_t^{\theta+\delta} \prod_{k=t}^{T-1} (\alpha(\mathbf{U} + \delta_\mathbf{U})^\top \mathbf{D}_{k+1}^{\theta+\delta} + \beta \mathbf{I}) \right) (\mathbf{v} + \delta_\mathbf{v}) \right] \right\|_2.
\end{aligned}
\tag{15}
$$

Let us define matrices $\mathcal{A}_t^\theta := \mathbf{D}_t^\theta \prod_{k=t}^{T-1}(\alpha \mathbf{U}^\top \mathbf{D}_{k+1}^\theta + \beta \mathbf{I})$ and similarly $\mathcal{A}_t^{\theta+\delta} := \mathbf{D}_t^{\theta+\delta} \prod_{k=t}^{T-1}(\alpha(\mathbf{U} + \delta_\mathbf{U})^\top \mathbf{D}_{k+1}^{\theta+\delta} + \beta \mathbf{I})$. Using this, we have,

$$
\begin{aligned}
&\left\| \frac{\partial L}{\partial \mathbf{W}}(\theta) - \frac{\partial L}{\partial \mathbf{W}}(\theta + \delta) \right\|_F \\
&= \alpha R_\mathbf{x} \left\| \sum_{t=0}^{T} \left[ c(\theta) \cdot \mathcal{A}_t^\theta \mathbf{v} - c(\theta + \delta) \cdot \mathcal{A}_t^{\theta+\delta}(\mathbf{v} + \delta_\mathbf{v}) \right] \right\|_2 \\
&\leq \alpha R_\mathbf{x} \left( |c(\theta) - c(\theta + \delta)| \cdot \left\| \sum_{t=0}^{T} \mathcal{A}_t^\theta \mathbf{v} \right\|_2 + \left\| \sum_{t=0}^{T} \mathcal{A}_t^\theta \mathbf{v} - \mathcal{A}_t^{\theta+\delta}(\mathbf{v} + \delta_\mathbf{v}) \right\|_2 \right) \\
&\leq \alpha R_\mathbf{x} \left( |c(\theta) - c(\theta + \delta)| \cdot \left\| \sum_{t=0}^{T} \mathcal{A}_t^\theta \mathbf{v} \right\|_2 + \left\| \sum_{t=0}^{T} (\mathcal{A}_t^\theta - \mathcal{A}_t^{\theta+\delta})\mathbf{v} \right\|_2 + \left\| \sum_{t=0}^{T} \mathcal{A}_t^{\theta+\delta}\delta_\mathbf{v} \right\|_2 \right) \\
&\leq \alpha R_\mathbf{x} \left( |c(\theta) - c(\theta + \delta)| \cdot R_\mathbf{v} \left\| \sum_{t=0}^{T} \mathcal{A}_t^\theta \right\|_2 + R_\mathbf{v} \left\| \sum_{t=0}^{T} \mathcal{A}_t^\theta - \mathcal{A}_t^{\theta+\delta} \right\|_2 + \|\delta_\mathbf{v}\|_2 \left\| \sum_{t=0}^{T} \mathcal{A}_t^{\theta+\delta} \right\|_2 \right). \tag{16}
\end{aligned}
$$

We will proceed by bounding the first term in the above equation. Consider,

$$
\begin{aligned}
\left\| \sum_{t=0}^{T} \mathcal{A}_t^\theta \right\|_2 &\leq \mathcal{D} \sum_{t=0}^{T} \left\| \prod_{k=t}^{T-1} (\alpha \mathbf{U}^\top \mathbf{D}_{k+1}^\theta + \beta \mathbf{I}) \right\|_2 \\
&\leq \mathcal{D} \sum_{t=0}^{T} (\alpha \mathcal{D} \cdot \|\mathbf{U}\|_2 + \beta)^{T-t} \\
&\leq \mathcal{D} \frac{|(\alpha \mathcal{D} \cdot \|\mathbf{U}\|_2 + \beta)^{T+1} - 1|}{|\alpha \mathcal{D} \cdot \|\mathbf{U}\|_2 + \beta - 1|} \\
&\overset{\varsigma_1}{\leq} \mathcal{D} \frac{(1 + \alpha \cdot (\mathcal{D}\|U\|_2 - 1))^{T+1} - 1}{\alpha |\mathcal{D}\|\mathbf{U}\|_2 - 1|} \\
&\overset{\varsigma_2}{\leq} 2\mathcal{D} \cdot (T + 1), \tag{17}
\end{aligned}
$$

where $\varsigma_1$ follows by setting $\beta = 1 - \alpha$ and $\varsigma_2$ follows by using the inequality $(1 + x)^r \leq 1 + 2rx$ for $(r - 1)x \leq 1/2$ and the fact that $\alpha \leq \frac{1}{4T \cdot |\mathcal{D}\|U\|_2 - 1|}$. Note that the third term in Equation (17) can be bounded in a similar way as above by $2\mathcal{D} \cdot (T + 1)$ using $\alpha \leq \frac{1}{4T \cdot R_\mathbf{U}}$. We now proceed to

bound the second term. Consider the following for any fixed value of $t$,

$$\left\|\mathcal{A}_t^\theta - \mathcal{A}_t^{\theta+\delta}\right\|_2 = \left\|\mathbf{D}_t^\theta \prod_{k=t}^{T-1}(\alpha\mathbf{U}^\top\mathbf{D}_{k+1}^\theta + \beta\mathbf{I}) - \mathbf{D}_t^{\theta+\delta}\prod_{k=t}^{T-1}(\alpha(\mathbf{U}+\delta_\mathbf{U})^\top\mathbf{D}_{k+1}^{\theta+\delta} + \beta\mathbf{I})\right\|_2$$

$$\leq \left\|(\mathbf{D}_t^\theta - \mathbf{D}_t^{\theta+\delta})\prod_{k=t}^{T-1}(\alpha\mathbf{U}^\top\mathbf{D}_{k+1}^\theta + \beta\mathbf{I})\right\|_2 + \mathcal{D}\left\|\prod_{k=t}^{T-1}(\alpha\mathbf{U}^\top\mathbf{D}_{k+1}^\theta + \beta\mathbf{I}) - \prod_{k=t}^{T-1}(\alpha(\mathbf{U}+\delta_\mathbf{U})^\top\mathbf{D}_{k+1}^{\theta+\delta} + \beta\mathbf{I})\right\|_2$$

$$\leq \left\|\mathbf{D}_t^\theta - \mathbf{D}_t^{\theta+\delta}\right\|_2 \cdot \left\|\prod_{k=t}^{T-1}(\alpha\mathbf{U}^\top\mathbf{D}_{k+1}^\theta + \beta\mathbf{I})\right\|_2 + \mathcal{D}\left\|\prod_{k=t}^{T-1}(\alpha\mathbf{U}^\top\mathbf{D}_{k+1}^\theta + \beta\mathbf{I}) - \prod_{k=t}^{T-1}(\alpha(\mathbf{U}+\delta_\mathbf{U})^\top\mathbf{D}_{k+1}^{\theta+\delta} + \beta\mathbf{I})\right\|_2$$

$$\leq \left\|\mathbf{D}_t^\theta - \mathbf{D}_t^{\theta+\delta}\right\|_2 \cdot (\alpha\|\mathbf{U}\|_2\mathcal{D} + \beta)^{T-t} + \mathcal{D}\underbrace{\left\|\prod_{k=t}^{T-1}(\alpha\mathbf{U}^\top\mathbf{D}_{k+1}^\theta + \beta\mathbf{I}) - \prod_{k=t}^{T-1}(\alpha(\mathbf{U}+\delta_\mathbf{U})^\top\mathbf{D}_{k+1}^{\theta+\delta} + \beta\mathbf{I})\right\|_2}_{(I)}.$$

Let $\Delta_k^\theta := \mathbf{D}_k^\theta - \mathbf{D}_k^{\theta+\delta}$. We will later show that $\left\|\Delta_k^\theta\right\|_2 \leq \Delta_\theta$ independent of the value of $k$. We focus on term $(I)$ in the expression above:

$$\left\|\prod_{k=t}^{T-1}(\alpha\mathbf{U}^\top\mathbf{D}_{k+1}^\theta + \beta\mathbf{I}) - \prod_{k=t}^{T-1}(\alpha(\mathbf{U}+\delta_\mathbf{U})^\top\mathbf{D}_{k+1}^{\theta+\delta} + \beta\mathbf{I})\right\|_2$$

$$\leq \left\|\prod_{k=t}^{T-1}(\alpha\mathbf{U}^\top\mathbf{D}_{k+1}^{\theta+\delta} + \beta\mathbf{I} + \alpha\mathbf{U}^\top\Delta_{k+1}^\theta) - \prod_{k=t}^{T-1}(\alpha\mathbf{U}^\top\mathbf{D}_{k+1}^{\theta+\delta} + \beta\mathbf{I} + \alpha\delta_\mathbf{U}^\top\mathbf{D}_{k+1}^{\theta+\delta})\right\|_2. \qquad (18)$$

Let $\mathcal{B}_k := \alpha\mathbf{U}^\top\mathbf{D}_k^{\theta+\delta} + \beta\mathbf{I}$, $\mathcal{C}_k := \alpha\mathbf{U}^\top\Delta_{k+1}^\theta$ and $\mathcal{G}_k := \alpha\delta_\mathbf{U}^\top\mathbf{D}_{k+1}^{\theta+\delta}$. Note that we have the following bounds on the operator norms of these matrices:

$$\|\mathcal{B}_k\|_2 \leq \alpha\mathcal{D}\cdot\|\mathbf{U}\|_2 + \beta = \mathcal{B}_{\max}, \quad \|\mathcal{C}_k\|_2 \leq \alpha\Delta_\theta\cdot\|\mathbf{U}\|_2 = \mathcal{C}_{\max}, \quad \|\mathcal{G}_k\| \leq \alpha\mathcal{D}\cdot\|\delta_\mathbf{U}\|_2 = \mathcal{G}_{\max}. \qquad (19)$$

By our assumptions on $\alpha$, $\mathcal{B}_k$ is invertible and $I + \mathcal{B}_k\mathcal{C}_k\mathcal{B}_k^{-1}$, $I + \mathcal{B}_k\mathcal{G}_k\mathcal{B}_k^{-1}$ are diagonalizable. Moreover, $\|\mathcal{B}_k^{-1}\| \leq 2\alpha\mathcal{D}\cdot\|\mathbf{U}\|_2 + \beta = \mathcal{B}_{\max}^{-1}$.

Hence, we can rewrite Equation (18) as,

$$\left\|\prod_{k=t}^{T-1}(\alpha\mathbf{U}^\top\mathbf{D}_{k+1}^{\theta+\delta} + \beta\mathbf{I}) - \prod_{k=t}^{T-1}(\alpha(\mathbf{U}+\delta_\mathbf{U})^\top\mathbf{D}_{k+1}^{\theta+\delta} + \beta\mathbf{I})\right\|_2$$

$$\leq \left\|\prod_{k=t}^{T-1}(\mathcal{B}_k + \mathcal{C}_k) - \prod_{k=t}^{T-1}(\mathcal{B}_k + \mathcal{G}_k)\right\|$$

$$\leq 4\|\mathcal{B}_t\|\cdot\left\|\prod_{k=t}^{T-1}\left(I + \mathcal{B}_t^{-1}\mathcal{C}_k\mathcal{B}_{t+1}\right) - \prod_{k=t}^{T-1}\left(I + \mathcal{B}_t^{-1}\mathcal{G}_k\mathcal{B}_{t+1}\right)\right\|$$

$$\leq 4\|\mathcal{B}_t\|\cdot\left((1 + \mathcal{B}_{\max}\cdot\mathcal{C}_{\max}\cdot\mathcal{B}_{\max}^{-1})^{T-t} - 1 + (1 + \mathcal{B}_{\max}\cdot\mathcal{G}_{\max}\cdot\mathcal{B}_{\max}^{-1})^{T-t} - 1\right), \qquad (20)$$

where $B_T := I$ and the last equation follows from the following fact: $\|\prod_{k=1}^T(I + C_k) - I\| \leq (\max_k\|C_k\| + 1)^T - 1$.

Combining the above term with Equation (16):

$$\left\|\sum_{t=0}^T \mathcal{A}_t^\theta - \mathcal{A}_t^{\theta+\delta}\right\|_2 \leq \sum_{t=0}^T\left\|\mathcal{A}_t^\theta - \mathcal{A}_t^{\theta+\delta}\right\|_2$$

$$\leq \Delta_\theta\cdot\sum_{t=0}^T(\alpha\mathcal{D}\cdot\|\mathbf{U}\|_2 + \beta)^{T-t} + \mathcal{D}\cdot\mathcal{B}_{\max}^{-1}\cdot\mathcal{B}_{\max}$$

$$\cdot\sum_{t=0}^T\left((1 + \mathcal{B}_{\max}\cdot\mathcal{C}_{\max}\cdot\mathcal{B}_{\max}^{-1})^{T-t} - 1 + (1 + \mathcal{B}_{\max}\cdot\mathcal{G}_{\max}\cdot\mathcal{B}_{\max}^{-1})^{T-t} - 1\right)$$

$$\leq \Delta_\theta\cdot\sum_{t=0}^T(\alpha\mathcal{D}\cdot\|\mathbf{U}\|_2 + \beta)^{T-t} + 2\mathcal{D}\cdot(\mathcal{B}_{\max}^{-1})^3\cdot(\mathcal{B}_{\max})^3\cdot T^2\cdot((\mathcal{C}_{\max})^2 + (\mathcal{G}_{\max})^2)$$

$$\overset{\zeta_1}{\leq} 2\Delta_\theta\cdot(T+1) + 2\mathcal{D}\cdot(\mathcal{B}_{\max}^{-1})^3\cdot(\mathcal{B}_{\max})^3\cdot T^2\cdot((\mathcal{C}_{\max})^2 + (\mathcal{G}_{\max})^2), \qquad (21)$$

where $\zeta_1$ follows by summing the geometric series and using the fact that $\alpha \leq \frac{1}{4T \cdot |\mathcal{D}\|U\|_2 - 1|}$.

Using the definition of $D_k^\theta = \operatorname{diag}(\sigma'(\mathbf{W}\mathbf{x}_k + \mathbf{U}\mathbf{h}_{k-1}))$ from Section 3 of the paper, we obtain a bound on $\Delta_k^\theta$.

$$
\left\| \mathbf{D}_k^\theta - \mathbf{D}_k^{\theta+\delta} \right\|_2 \leq 2 \left( R_\mathbf{x} \cdot \|\delta_\mathbf{W}\|_2 + \sqrt{\hat{D}} \cdot \|\delta_\mathbf{U}\|_2 + R_\mathbf{U} \cdot \|\mathbf{h}_{k-1} - \mathbf{h}'_{k-1}\|_2 \right)
$$
$$
\overset{\zeta_1}{\leq} 2 \left( R_\mathbf{x} \cdot \|\delta_\mathbf{W}\|_2 + \sqrt{\hat{D}} \cdot \|\delta_\mathbf{U}\|_2 + R_\mathbf{U} \cdot \frac{2\sqrt{\hat{D}} \cdot \|\delta_\mathbf{U}\|_2 + 2\|\delta_\mathbf{W}\|_2 R_\mathbf{x}}{\|\|U\|_2 - 1\|} \right), \quad (22)
$$

where $\zeta_1$ follows from using the bound from Equation (12). Combining bounds obtained in Equations (16), (17), (21) and (22), we obtain that,

$$
\left\| \frac{\partial L}{\partial \mathbf{W}}(\theta) - \frac{\partial L}{\partial \mathbf{W}}(\theta + \delta) \right\|_F \leq \mathcal{O}(\alpha T) \cdot \|\delta\|_F, \quad \text{for}
$$
$$
\alpha \leq \min \left( \frac{1}{4T \cdot |\mathcal{D}\|U\|_2 - 1|}, \frac{1}{4T \cdot R_\mathbf{U}}, \frac{1}{2T \cdot \Delta_\theta \|U\|_2}, \frac{1}{T \cdot |\|\mathbf{U}\|_2 - 1|} \right)
$$

where the $\mathcal{O}$ notation hides *polynomial* dependence of the Lipschitz smoothness constant of $L$ on $R_\mathbf{W}, R_\mathbf{U}, R_\mathbf{v}, R_\mathbf{x}, \|\mathbf{U}\|_2, \|\mathbf{W}\|_2$ and the ambient dimensions $D, \hat{D}$.

**Deviation bound for $\frac{\partial L}{\partial \mathbf{U}}$**: Following similar arguments as we did above for $\frac{\partial L}{\partial \mathbf{W}}$, we can derive the perturbation bound for the term $\frac{\partial L}{\partial \mathbf{U}}$ as

$$
\left\| \frac{\partial L}{\partial \mathbf{U}}(\theta) - \frac{\partial L}{\partial \mathbf{U}}(\theta + \delta) \right\|_F = \mathcal{O}(\alpha T) \cdot \|\delta\|_F \quad (23)
$$

where the $\mathcal{O}$ notation is the same as above.

Using our bounds in corollary 2.2 of [15], we obtain the following convergence theorem.

**Theorem 3.1** (Convergence Bound). *Let $[(\mathbf{X}_1, y_1), \ldots, (\mathbf{X}_n, y_n)]$ be the given labeled sequential training data. Let $L(\theta) = \frac{1}{n} \sum_i L(\mathbf{X}_i, y_i; \theta)$ be the loss function with $\theta = (\mathbf{W}, \mathbf{U}, \mathbf{v})$ be the parameters of FastRNN architecture (2) with $\beta = 1 - \alpha$ and $\alpha$ such that*

$$
\alpha \leq \min \left( \frac{1}{4T \cdot |\mathcal{D}\|\mathbf{U}\|_2 - 1|}, \frac{1}{4T \cdot R_\mathbf{U}}, \frac{1}{2T \cdot \Delta_\theta \|\mathbf{U}\|_2}, \frac{1}{T \cdot |\|\mathbf{U}\|_2 - 1|} \right),
$$

*where $\mathcal{D} = \sup_{\theta,k} \|\mathbf{D}_k^\theta\|_2$. Then, randomized stochastic gradient descent [15], a minor variation of SGD, when applied to the data for a maximum of $M$ iteration outputs a solution $\hat{\theta}$ such that:*

$$
\mathbb{E}[\|\nabla_\theta L(\hat{\theta})\|_2^2\|] \leq \mathcal{B}_M := \frac{\mathcal{O}(\alpha T) L(\theta_0)}{M} + \left( \bar{D} + \frac{4R_\mathbf{W} R_\mathbf{U} R_\mathbf{v}}{\bar{D}} \right) \frac{\mathcal{O}(\alpha T)}{\sqrt{M}},
$$

*where $R_\mathbf{X} = \max_\mathbf{X} \|\mathbf{X}\|_F$ for $\mathbf{X} = \{\mathbf{U}, \mathbf{W}, \mathbf{v}\}$, $L(\theta_0)$ is the loss of the initial classifier, and the step-size of the $k$-th SGD iteration is fixed as: $\gamma_k = \min \left\{ \frac{1}{\mathcal{O}(\alpha T)}, \frac{\bar{D}}{T\sqrt{M}} \right\}, k \in [M], \ \bar{D} \geq 0$.*

## A.1 Generalization Bound for FastRNN

In this subsection, we compute the Rademacher complexity of the class of real valued scalar gated recurrent neural networks such that $\|\mathbf{U}\|_F \leq R_\mathbf{U}, \|\mathbf{W}\|_F \leq R_\mathbf{W}$. Also the input $\mathbf{x}_t$ at time step $t$ is assumed to be point-wise bounded $\|\mathbf{x}_t\|_2 \leq R_\mathbf{x}$ The update equation of FastRNN is given by

$$
\mathbf{h}_t = \alpha\sigma(\mathbf{W}\mathbf{x}_t + \mathbf{U}\mathbf{h}_{t-1}) + \beta\mathbf{h}_{t-1}.
$$

For the purpose of this section, we use the shorthand $\mathbf{h}_t^i$ to denote the hidden vector at time $t$ corresponding to the $i^{th}$ data point $\mathbf{X}^i$. We denote the Rademacher complexity of a $T$ layer FastRNN

by $\mathcal{R}_n(\mathcal{F}_T)$ evaluated using $n$ data points.

$$n\mathcal{R}_n(\mathcal{F}_T) = \mathbb{E}_\epsilon \left[ \sup_{\mathbf{W},\mathbf{U}} \left\| \sum_{i=1}^{n} \epsilon_i \mathbf{h}_T^i \right\| \right]$$

$$= \mathbb{E}_\epsilon \left[ \sup_{\mathbf{W},\mathbf{U}} \left\| \sum_{i=1}^{n} \epsilon_i \left( \alpha\sigma(\mathbf{W}\mathbf{x}_T^i + \mathbf{U}\mathbf{h}_{T-1}^i) + \beta\mathbf{h}_{T-1}^i \right) \right\| \right]$$

$$\overset{\zeta_1}{\leq} \mathbb{E}_\epsilon \left[ \sup_{\mathbf{W},\mathbf{U}} \beta \left\| \sum_{i=1}^{n} \epsilon_i \mathbf{h}_{T-1}^i \right\| \right] + \mathbb{E}_\epsilon \left[ \sup_{\mathbf{W},\mathbf{U}} \alpha \left\| \sum_{i=1}^{n} \epsilon_i(\sigma(\mathbf{W}\mathbf{x}_T^i + \mathbf{U}\mathbf{h}_{T-1}^i)) \right\| \right]$$

$$\overset{\zeta_2}{\leq} \beta\mathcal{R}_n(\mathcal{F}_{T-1}) + 2\mathbb{E}_\epsilon \left[ \sup_{\mathbf{W},\mathbf{U}} \alpha \left\| \sum_{i=1}^{n} \epsilon_i(\mathbf{W}\mathbf{x}_T^i + \mathbf{U}\mathbf{h}_{T-1}^i) \right\| \right]$$

$$\overset{\zeta_3}{\leq} \beta\mathcal{R}_n(\mathcal{F}_{T-1}) + 2\alpha\mathbb{E}_\epsilon \left[ \sup_{W} \left\| \sum_{i=1}^{n} \epsilon_i \mathbf{W}\mathbf{x}_T^i \right\| \right] + 2\alpha\mathbb{E}_\epsilon \left[ \sup_{\mathbf{W},\mathbf{U}} \left\| \sum_{i=1}^{n} \epsilon_i \mathbf{U}\mathbf{h}_{T-1}^i) \right\| \right]$$

$$\overset{\zeta_4}{\leq} \beta\mathcal{R}_n(\mathcal{F}_{T-1}) + 2\alpha R_\mathbf{W}\mathbb{E}_\epsilon \left[ \left\| \sum_{i=1}^{n} \epsilon_i \mathbf{x}_T^i \right\| \right] + 2\alpha R_\mathbf{U}\mathbb{E}_\epsilon \left[ \sup_{\mathbf{W},\mathbf{U}} \left\| \sum_{i=1}^{n} \epsilon_i \mathbf{h}_{T-1}^i \right\| \right]$$

$$\leq (\beta + 2\alpha R_\mathbf{U})\mathcal{R}_n(\mathcal{F}_{T-1}) + 2\alpha R_\mathbf{W} R_\mathbf{X} \sqrt{n}$$

$$\leq (\beta + 2\alpha R_\mathbf{U})^2 \mathcal{R}_n(\mathcal{F}_{T-2}) + 2\alpha R_\mathbf{W} R_\mathbf{X} \sqrt{n}(1 + (\beta + 2\alpha R_\mathbf{U}))$$

$$\vdots$$

$$\leq 2\alpha R_\mathbf{W} R_\mathbf{X} \sum_{t=0}^{T-1} (\beta + 2\alpha R_\mathbf{U})^{T-t} \sqrt{n}$$

$$\leq 2\alpha R_\mathbf{W} R_\mathbf{X} \left( \frac{(\beta + 2\alpha R_\mathbf{U})^{T+1} - 1}{(\beta + 2\alpha R_\mathbf{U}) - 1} \right) \sqrt{n}$$

$$\leq 2\alpha R_\mathbf{W} R_\mathbf{X} \left( \frac{(1 + \alpha(2R_U - 1))^{T+1} - 1}{\alpha(2R_\mathbf{U} - 1)} \right) \sqrt{n}$$

$$\overset{\zeta_5}{\leq} 2R_\mathbf{W} R_\mathbf{X} \left( \frac{2\alpha(2R_\mathbf{U} - 1)(T + 1)}{(2R_\mathbf{U} - 1)} \right) \sqrt{n},$$

where $\zeta_1, \zeta_3$ follows by triangle inequality and noting that the terms in the sum of expectation are pointwise bigger than the previous term, $\zeta_2$ follows from the Ledoux-Talagrand contraction, $\zeta_4$ follows using an argument similar from Lemma 1 in [16] and $\zeta_5$ holds for $\alpha \leq \frac{1}{2(2R_\mathbf{U}-1)T}$.

**Theorem 3.2** (Generalization Error Bound). *[6] Let $\mathcal{Y}, \hat{\mathcal{Y}} \subseteq [0,1]$ and let $\mathcal{F}_T$ denote the class of FastRNN with $\|\mathbf{U}\|_F \leq R_\mathbf{U}, \|\mathbf{W}\|_F \leq R_\mathbf{W}$. Let the final classifier be given by $\sigma(\mathbf{v}^\top \mathbf{h}_T)$, $\|\mathbf{v}\|_2 \leq R_\mathbf{v}$. Let $L : \mathcal{Y} \times \hat{\mathcal{Y}} \to [0, B]$ be any 1-Lipschitz loss function. Let $D$ be any distribution on $\mathcal{X} \times \mathcal{Y}$ such that $\|\mathbf{x}_{it}\|_2 \leq R_x$ a.s. Let $0 \leq \delta \leq 1$. For all $\beta = 1 - \alpha$ and $alpha$ such that,*

$$\alpha \leq \min\left( \frac{1}{4T \cdot |\mathcal{D}\|\mathbf{U}\|_2 - 1|}, \frac{1}{4T \cdot R_\mathbf{U}}, \frac{1}{T \cdot |\|\mathbf{U}\|_2 - 1|} \right).$$

*where $\mathcal{D} = \sup_{\theta,k} \|\mathbf{D}_k^\theta\|_2$, we have that with probability at least $1 - \delta$, all functions $f \in \mathbf{v} \circ \mathcal{F}_T$ satisfy,*

$$\mathbb{E}_D[L(f(\mathbf{X}), y)] \leq \frac{1}{n} \sum_{i=1}^{n} L(f(\mathbf{X}_i), y_i) + \mathcal{C}\frac{O(\alpha T)}{\sqrt{n}} + B\sqrt{\frac{\ln(\frac{1}{\delta})}{n}},$$

*where $\mathcal{C} = R_\mathbf{W} R_\mathbf{U} R_\mathbf{x} R_\mathbf{v}$ represents the boundedness of the parameter matrices and the data.*

The Rademacher complexity bounds for the function class $\mathcal{F}_T$ have been instantiated from the calculations above.

# B  Dataset Information

**Google-12 & Google-30:** Google Speech Commands dataset contains 1 second long utterances of 30 short words (30 classes) sampled at 16KHz. Standard log Mel-filter-bank featurization with 32 filters over a window size of 25ms and stride of 10ms gave 99 timesteps of 32 filter responses for a 1-second audio clip. For the 12 class version, 10 classes used in Kaggle's Tensorflow Speech Recognition challenge[1] were used and remaining two classes were noise and background sounds (taken randomly from remaining 20 short word utterances). Both the datasets were zero mean - unit variance normalized during training and prediction.

**Wakeword-2:** Wakeword-2 consists of 1.63 second long utterances sampled at 16KHz. This dataset was featurized in the same way as the Google Speech Commands dataset and led to 162 timesteps of 32 filter responses. The dataset was zero mean - unit variance normalized during training and prediction.

**HAR-2[2]:** Human Activity Recognition (HAR) dataset was collected from an accelerometer and gyroscope on a Samsung Galaxy S3 smartphone. The features available on the repository were directly used for experiments. The 6 activities were merged to get the binarized version. The classes {Sitting, Laying, Walking_Upstairs} and {Standing, Walking, Walking_Downstairs} were merged to obtain the two classes. The dataset was zero mean - unit variance normalized during training and prediction.

**DSA-19[3]:** This dataset is based on Daily and Sports Activity (DSA) detection from a resource-constrained IoT wearable device with 5 Xsens MTx sensors having accelerometers, gyroscopes and magnetometers on the torso and four limbs. The features available on the repository were used for experiments. The dataset was zero mean - unit variance normalized during training and prediction.

**Yelp-5:** Sentiment Classification dataset based on the text reviews[4]. The data consists of 500,000 train points and 500,000 test points from the first 1 million reviews. Each review was clipped or padded to be 300 words long. The vocabulary consisted of 20000 words and 128 dimensional word embeddings were jointly trained with the network.

**Penn Treebank:** 300 length word sequences were used for word level language modeling task using Penn Treebank (PTB) corpus. The vocabulary consisted of 10,000 words and the size of trainable word embeddings was kept the same as the number of hidden units of architecture.

**Pixel-MNIST-10:** Pixel-by-pixel version of the standard MNIST-10 dataset [5]. The dataset was zero mean - unit variance normalized during training and prediction.

**AmazonCat-13K [34, 8]:** AmazonCat-13K is an extreme multi-label classification dataset with 13,330 labels. The raw text from title and content for Amazon products was provided as an input with each product being assigned to multiple categories. The input text was clipped or padded to ensure that it was 500 words long with a vocabulary of size 267,134. The 50 dimensional trainable word embeddings were initialized with GloVe vectors trained on Wikipedia.

## Evaluation on Multilabel Dataset

The models were trained on the AmazonCat-13K dataset using Adam optimizer with a learning rate of 0.009 and batch size of 128. Binary Cross Entropy loss was used where the output of each neuron corresponds to the probability of a label being positive. 128 hidden units were chosen across architectures and were trained using PyTorch framework.

The results in Table 6 show that FastGRNN-LSQ achieves classification performance similar to state-of-the-art gated architectures (GRU, LSTM) while still having 2-3x lower memory footprint. Note that the model size reported doesn't include the embeddings and the final linear classifier which are memory intensive when compared to the model itself. FastRNN, as shown in the earlier

Table 6: Extreme Multi Label Classification

| Dataset | AmazonCat - 13K | | | | | |
|---|---|---|---|---|---|---|
| | P@1 | P@2 | P@3 | P@4 | P@5 | Model Size - RNN (KB) |
| GRU | 92.82 | 85.18 | 77.09 | 69.42 | 61.85 | 268 |
| RNN | 40.24 | 28.13 | 22.83 | 20.29 | 18.25 | 89.5 |
| FastGRNN-LSQ | 92.66 | 84.67 | 76.19 | 66.67 | 60.63 | 90.0 |
| FastRNN | 91.03 | 81.75 | 72.37 | 64.13 | 56.81 | 89.5 |
| UGRNN | 92.84 | 84.93 | 76.33 | 68.27 | 60.63 | 179 |

experiments, stabilizes standard RNN and achieves an improvement of over 50% in classification accuracy (P@1).

## C  Supplementary Experiments

**Accuracy vs Model Size:** This paper evaluates the trade-off between model size (in the range 0-128Kb) and accuracy across various architectures.

Figure 3: Accuracy vs Model Size

Figure 4: Accuracy vs Model Size

Figures 3, 4 show the plots for analyzing the model-size vs accuracy trade-off for FastGRNN, FastGRNN-LSQ along with leading unitary method SpectralRNN and the gated methods like UGRNN, GRU, and LSTM. FastGRNN is able to achieve state-of-the-art accuracies on Google-12 and Google-30 datasets at significantly lower model sizes as compared to other baseline methods.

**Bias due to the initial hidden states:** In order to understand the bias induced at the output by the initial hidden state $h_0$, we evaluated a trained FastRNN classifier on the Google-12 dataset with 3 different initializations sampled from a standard normal distribution. The resulting accuracies had a mean value of $92.08$ with a standard deviation of $0.09$, indicating that the initial state does not induce a bias in FastRNN prediction in the learning setting. In the non-learning setting, the initial

state can bias the final solution for very small values of $\alpha$. Indeed, setting $\alpha = 0$ and $\beta = 1$ will bias the final output to the initial state. However, as Figure 5 indicates, such an effect is observed only for extremely small values of $\alpha \in (0, 0.005)$. In addition, there is a large enough range for $\alpha \in (0.005, 0.08)$ where the final output of FastRNN is not biased and is easily learnt by FastRNN.

Figure 5: Accuracy vs $\alpha$ in non-learning setting where the parameters of the classifier was learnt and evaluated for a range of fixed $\alpha$ values (using 99 timesteps).

$\alpha$ **and** $\beta$ **of FastRNN:** $\alpha$ and $\beta$ are the trainable weights of the residual connection in FastRNN. Section 3.1.1 shows that FastRNN has provably stable training for the setting of $\alpha/\beta = O(1/T)$. Table 7 shows the learnt values of $\alpha$ and $\beta$ for various timesteps ($T$) across 3 datasets.

Table 7: Scaling of $\alpha$ and $\beta$ vs Timesteps for FastRNN with $\tanh$ non-linearity: With $\alpha$ set as a trainable parameter, it scales as $O(1/T)$ with the number of timesteps as suggested by Theorem 3.1.

| Google-12 | | | HAR-2 | | | MNIST-10 | | |
|---|---|---|---|---|---|---|---|---|
| Timesteps | $\alpha$ | $\beta$ | Timesteps | $\alpha$ | $\beta$ | Timesteps | $\alpha$ | $\beta$ |
| 99 | 0.0654 | 0.9531 | 128 | 0.0643 | 0.9652 | 112 | 0.0617 | 0.9447 |
| 33 | 0.2042 | 0.8898 | 64 | 0.1170 | 0.9505 | 56 | 0.1193 | 0.9266 |
| 11 | 0.5319 | 0.7885 | 32 | 0.1641 | 0.9606 | 28 | 0.2338 | 0.8746 |
| 9 | 0.5996 | 0.7926 | 16 | 0.2505 | 0.9718 | 14 | 0.3850 | 0.8251 |
| 3 | 0.6878 | 0.8246 | 8 | 0.3618 | 0.9678 | 7 | 0.5587 | 0.8935 |

## D    Compression Components of FastGRNN

The Compression aspect of FastGRNN has 3 major components: 1) Low-rank parameterization (**L**) 2) Sparsity (**S**) and 3) Byte Quantization (**Q**). The general trend observed across dataset is that low-rank parameterization increase classification accuracies while the sparsity and quantization help reduced the model sizes by 2x and 4x respectively across datasets.

Tables 8, 9 and 10 show the trend when each of the component is gradually removed from FastGRNN to get to FastGRNN-LSQ. Note that the hyperparameters have been re-tuned along with the relevant constraints to obtain each model in the table. Figure 6 shows the effect of each of LSQ components for two Google datasets.

## E    Hyperparameters of FastGRNN for reproducibility:

Table 11 lists the hyperparameters which were used to run the experiments with a random-seed of 42 on a P40 GPU card with CUDA 9.0 and CuDNN 7.1. One can use the Piece-wise linear approximations of $\tanh$ or sigmoid if they wish to quantize the weights.

## F    Timing Experiments on more IoT boards

Table 12 summarizes the timing results on the Raspberry Pi which has a more powerful processor as compared with Arduino Due. Note that the Raspberry Pi has special instructions for floating point

Table 8: Components of Compression

| Dataset | FastGRNN | | FastGRNN-Q | | FastGRNN-SQ | | FastGRNN-LSQ | |
|---|---|---|---|---|---|---|---|---|
| | Accuracy (%) | Model Size (KB) | Accuracy (%) | Model Size (KB) | Accuracy (%) | Model Size (KB) | Accuracy (%) | Model Size (KB) |
| Google-12 | 92.10 | 5.50 | 92.60 | 22 | 93.76 | 41 | 93.18 | 57 |
| Google-30 | 90.78 | 6.25 | 91.18 | 25 | 91.99 | 38 | 92.03 | 45 |
| HAR-2 | 95.59 | 3.00 | 96.37 | 17 | 96.81 | 28 | 95.38 | 29 |
| DSA-19 | 83.73 | 3.25 | 83.93 | 13 | 85.67 | 22 | 85.00 | 208 |
| Yelp-5 | 59.43 | 8.00 | 59.61 | 30 | 60.52 | 130 | 59.51 | 130 |
| Pixel-MNIST-10 | 98.20 | 6.00 | 98.58 | 25 | 98.72 | 37 | 98.72 | 71 |

Table 9: Components of Compression for Wakeword-2

| Dataset | FastGRNN | | FastGRNN-Q | | FastGRNN-SQ | | FastGRNN-LSQ | |
|---|---|---|---|---|---|---|---|---|
| | F1 Score | Model Size (KB) | F1 Score | Model Size (KB) | F1 Score | Model Size (KB) | F1 Score | Model Size (KB) |
| Wakeword-2 | 97.83 | 1 | 98.07 | 4 | 98.27 | 8 | 98.19 | 8 |

Table 10: Components of Compression for PTB

| Dataset | FastGRNN | | FastGRNN-Q | | FastGRNN-SQ | | FastGRNN-LSQ | |
|---|---|---|---|---|---|---|---|---|
| | Test Perplexity | Model Size (KB) | Test Perplexity | Model Size (KB) | Test Perplexity | Model Size (KB) | Test Perplexity | Model Size (KB) |
| PTB-10000 | 116.11 | 38.5 | 115.71 | 154 | 115.23 | 384 | 115.92 | 513 |

Figure 6: Figures (a) and (b) show the effect of LSQ components over the model size range of 0-64KB.

arithmetic and hence quantization doesn't provide any benefit with respect to compute in this case, apart from bringing down the model size considerably.

## G  Vectorized FastRNN

As a natural extension of FastRNN, this paper also benchmarked FastRNN-vector wherein the scalar $\alpha$ in FastRNN was extended to a vector and $\beta$ was substituted with $\zeta(1 - \alpha) + \nu$ with $\zeta$ and $\nu$ are trainable scalars in $[0, 1]$. Tables 13, 14 and 15 summarize the results for FastRNN-vector and a direct comparison shows that the gating enable FastGRNN is more accurate than FastRNN-vector. FastRNN-vector used tanh as the non-linearity in most cases except for a few (indicated by [+]) where ReLU gave slightly better results.

Table 11: Hyperparameters for reproducibility - FastGRNN-Q

| Dataset | Hidden Units | $r_w$ | $r_u$ | $s_w$ | $s_u$ | Nonlinearity | Optimizer |
|---|---|---|---|---|---|---|---|
| Google-12 | 100 | 16 | 25 | 0.30 | 0.30 | sigmoid | Momentum |
| Google-30 | 100 | 16 | 35 | 0.20 | 0.20 | tanh | Momentum |
| Wakeword-2 | 32 | 10 | 15 | 0.20 | 0.30 | tanh | Momentum |
| Yelp-5 | 128 | 16 | 32 | 0.30 | 0.30 | sigmoid | Adam |
| HAR-2 | 80 | 5 | 40 | 0.20 | 0.30 | tanh | Momentum |
| DSA-19 | 64 | 16 | 20 | 0.15 | 0.05 | sigmoid | Adam |
| Pixel-MNIST-10 | 128 | 1 | 30 | 1.00 | 0.30 | sigmoid | Adam |
| PTB-10000 | 256 | 64 | 64 | 0.30 | 0.30 | sigmoid | Adam |

Table 12: Prediction Time on Raspberry Pi 3 (ms)

| Method | Google-12 | HAR-2 | Wakeword-2 |
|---|---|---|---|
| FastGRNN | 7.7 | 1.8 | 2.5 |
| RNN | 15.7 | 2.9 | 3.6 |
| UGRNN | 29.7 | 5.6 | 9.5 |
| SpectralRNN | 123.2 | 391.0 | 17.2 |

Table 13: FastRNN Vector - 1

| Dataset | Accuracy (%) | Model Size (KB) | Train Time (hr) |
|---|---|---|---|
| Google-12 | 92.98[+] | 57 | 0.71 |
| Google-30 | 91.68[+] | 64 | 1.63 |
| HAR-2 | 95.24[+] | 19 | 0.06 |
| DSA-19 | 83.24 | 322 | 0.04 |
| Yelp-5 | 57.19 | 130 | 3.73 |
| Pixel-MNIST-10 | 97.27 | 44 | 13.75 |

Table 14: FastRNN Vector - 2

| Dataset | F1 Score | Model Size (KB) | Train Time (hr) |
|---|---|---|---|
| Wakeword-2 | 97.82 | 8 | 0.86 |

Table 15: FastRNN Vector - 3

| Dataset | Test Perplexity | Train Perplexity | Model Size (KB) | Train Time (min) |
|---|---|---|---|---|
| PTB-300 | 126.84 | 98.29 | 513 | 11.7 |

## H  Effects of Regularization for Language Modeling Tasks

This section studies the effect of various regularizations for Language Modeling tasks with the PTB dataset. [36] achieved state-of-the-art performance on the PTB dataset using a variety of different regularizations and this sections combines those techniqeus with FastGRNN and FastGRNN-LSQ. Table 16 summarizes the train and test perplexity of FastGRNN. The addition of an extra layer leads to a reduction of 10 points on the test perplexity score as compared to a single layer architecture of FastGRNN. Other regularizations like weight decay and weight dropping also lead to gains of upto 8 points in test perplexity as compared to the baseline FastGRNN architecture, exhibiting that such regularization techniques can be combined with the proposed architectures to obtain better dataset specific performance, especially on the language modelling tasks of the PTB dataset.

The experiments carried out in this paper on the PTB dataset use a sequence length of 300 as compared to those used in [38, 53, 21, 35, 36] which are generally in the range of 35-70. While standard recurrent architectures are known to work with such short sequence lengths, they typically exhibit unstable behavior in the regime where the sequence lengths are longer. These experiments exhibit the stability properties of FastGRNN (with 256 hidden units) in this regime of long sequence lengths with limited compute and memory resources.

Table 16: Language Modeling on PTB - Effect of regularization on FastGRNN

| Method | Hidden Units | Test Perplexity | Train Perplexity |
|---|---|---|---|
| 1-layer | 256 | 116.11 | 81.31 |
| 2-layer | 256 | 106.23 | 69.37 |
| 1-layer + Weight decay | 256 | 111.57 | 76.89 |
| 1-layer + Weight-dropping | 256 | 108.56 | 72.46 |
| 1-layer + AR/TAR | 256 | 112.78 | 78.79 |

## Footnotes

[1] https://www.kaggle.com/c/tensorflow-speech-recognition-challenge

[2] https://archive.ics.uci.edu/ml/datasets/human+activity+recognition+using+smartphones

[3] https://archive.ics.uci.edu/ml/datasets/Daily+and+Sports+Activities

[4] https://www.yelp.com/dataset/challenge

[5] http://yann.lecun.com/exdb/mnist/