[Reviews · NeurIPS 2018]

Reviewer 1
This paper presents an adaptation to the peep-hole connection from [22] to create fast, accurate, and, small RNNs. There are two variants introduced: (1) FastRNN which adds two extra learnable parameters to the vanilla RNN to regulate the computation flow between the hidden state and the nonlinear mapping of inputs and states (2) FastGRNN which replaces the two parameters with gating functions which share input/state matrices with the nonlinear projection but have separate biases. Furthermore, the FastGRNN utilises low-rank sparse representation for matrices with constraints that helps with compressing the size of the model. Theoretical analysis studies the convergence and stability of these models vs normal RNNs while the extensive experimental setup shows that these models are capable of achieving comparable results to state-of-the-art or comparable models (e.g. LSTMs) with much smaller network sizes. I really like the paper and the results reported and I'm sure this is going to have a great impact in the field. I can suggest a few improvements that could make the paper even better: - While the performance of these models have been studied in comparison to competing RNN formulations, the effects of regularisation have not been analysed. For example, the results reported on PTB are far worse than where the SOTA is and recent results on those tasks show that heavy regularisation of LSTMs for example can result in massive improvements in performance ("Regularizing and Optimizing LSTM Language Models", Merity et al 2017). So how would these models behave under heavy regularisation if one wants to attain high performances? - Also there are other work in this area such as "Quasi-Recurrent Neural Networks", Bradbury et al 2017 which also try to provide faster alternatives to LSTMs. How does FastGRNN perform against these methods? - While the balance between \alpha and \beta has been discussed briefly in the paper, it would have been better if we had further experimental results on them. Fig 4 in appendix is a good start, but what about \beta, how does that behave? Do we see often that \beta = 1 - \alpha as speculated in the paper? How does the ratio \alpha / \beta change with sequence length or different datasets? There are many interesting results here that would be good to have a report on - The paper has considerable redundancy in the introduction Sec 1 and related work Sec 2 which can be condensed. This will provide space to bring forward some of the interesting findings and results from appendix such as Fig 4 which.
Reviewer 2
This work presents a variant of recurrent neural networks, including its gated version. The key idea is a so-called "peephole" connection which mixes the previous hidden unit with the network hidden output. The proposed method is well supported by theoretical findings and good empirical evidences. The resulting networks are very compact and can be used in embedding products. Several things need to be clarified: 1) It is unclear how large is the learned beta. Figure 4 in appendex only gives the alpha results. How did you set beta? Was it close to 1-alpha? 2) In the non-learning setting, alpha>0 and 0
Reviewer 3
In this work, the authors propose a Recurrent Neural Network variant which is both accurate and efficient. FastRNN develops a leaky integrator unit inspired peephole connection which has only two extra scalar parameters. FastGRNN then extends the peephole to a gated architecture by reusing the RNN matrices in the gate to match state-of-the-art accuracies but with a 2-4x smaller model. And after the low-rank sparse and quantized approximation, FastGRNN could make more accurate predictions with up to a 35x smaller model as compared to leading unitary and gated RNN techniques! Overall, the accuracy and the huge model size reduction ratio is very impressive. The paper gives a very clear review on the various RNN models and illustrate their proposed approach very clearly. I am not an expert on the model size pruning area but from educated guess, the experiment results are extensive, solid and impressive. And it’s appealing that the model is actually evaluated on embedded platform like Arduino and Raspberry Pi.